# Collider bias correction for multiple covariates in GWAS using robust multivariable Mendelian randomization

**Peiyao Wang[1], Zhaotong Lin[1,2], Haoran Xue[1,3], Wei Pan[1] ***

1 Division of Biostatistics and Health Data Science, University of Minnesota, Minneapolis, Minnesota, United States of America, 2 Department of Statistics, Florida State University, Tallahassee, Florida, United States of America, 3 Department of Biostatistics, City University of Hong Kong, Hong Kong, China

* panxx014@umn.edu

**Data Availability Statement:** One needs to apply to UK Biobank (https://www.ukbiobank.ac.uk/) for approval to access the individual-level data used here. All public datasets used or obtained in our simulations and real data analyses are available

## Abstract

Genome-wide association studies (GWAS) have identified many genetic loci associated with complex traits and diseases in the past 20 years. Multiple heritable covariates may be added into GWAS regression models to estimate direct effects of genetic variants on a focal trait, or to improve the power by accounting for environmental effects and other sources of trait variations. When one or more covariates are causally affected by both genetic variants and hidden confounders, adjusting for them in GWAS will produce biased estimation of SNP effects, known as collider bias. Several approaches have been developed to correct collider bias through estimating the bias by Mendelian randomization (MR). However, these methods work for only one covariate, some of which utilize MR methods with relatively strong assumptions, both of which may not hold in practice. In this paper, we extend the bias-correction approaches in two aspects: first we derive an analytical expression for the collider bias in the presence of multiple covariates, then we propose estimating the bias using a robust multivariable MR (MVMR) method based on constrained maximum likelihood (called MVMR-cML), allowing the presence of invalid instrumental variables (IVs) and correlated pleiotropy. We also established the estimation consistency and asymptotic normality of the new bias-corrected estimator. We conducted simulations to show that all methods mitigated collider bias under various scenarios. In real data analyses, we applied the methods to two GWAS examples, the first a GWAS of waist-hip ratio with adjustment for only one covariate, body-mass index (BMI), and the second a GWAS of BMI adjusting metabolomic principle components as multiple covariates, illustrating the effectiveness of bias correction.

## Author summary

Genome-wide association studies (GWAS) are powerful in identifying genetic variants influencing complex traits and diseases. However, adjusting for heritable covariates in GWAS may introduce collider bias when both genetic variants and confounders may causally influence these covariates. In this study, for the first time we derived the analytical form of the bias term in GWAS with multiple covariates, enabling bias estimation and

from the Zendo repository (https://doi.org/10.5281/zenodo.10947055). The code to reproduce the results of this study is available at https://github.com/peiyao2017/MV-cML-bias-Adjustment.

**Funding:** This research was supported by NIH grants R01 AG065636 (to PW, ZL, HX, WP), R01 AG069895 (WP), RF1 AG067924 (WP), U01 AG073079 (PW, WP), R01 HL116720 (WP), and R01 GM126002 (WP). The funders had no role in the study design, data collection and analysis, decision to publish, or preparation of the manuscript.

**Competing interests:** The authors have declared that no competing interests exist.

correction using any MVMR method. On the other hand, many existing MVMR methods may not be robust to invalid IVs and are designed for independent samples. Since GWAS data of multiple traits are needed, overlapping samples become inevitable. Hence, while investigating the performance of many MVMR methods, we mainly adopt MVMR-cML, a novel MVMR approach robust to invalid IVs and sample overlap. Our simulations underscore that most MVMR methods effectively reduce collider bias across various scenarios. Furthermore, by accounting for correlations among GWAS statistics, as well as the linkage disequilibrium (LD) between the target SNP and IVs, we establish the consistency and asymptotic normality of the bias-corrected estimator based on MVMR-cML. The application of our bias-correction approach to two published GWAS data examples illustrates its utility and efficacy.

## Introduction

Genome-wide association studies (GWAS) have revolutionized genetics of complex diseases and traits by identifying many novel associations [1]. This breakthrough has not only paved the way for the development of new therapeutics but has also enabled early prevention strategies [2]. A notable example is a mega-GWAS conducted in 2014, which revealed 108 loci linked to schizophrenia, providing valuable insights for the development of novel drugs [3]. Moreover, the publication of GWAS results has facilitated secondary genetic epidemiological analyses, such as Mendelian randomization (MR) [4]. Despite these advancements, some limitations of GWAS remain.

Trait-associated single nucleotide polymorphisms (SNPs) detected through GWAS can only account for a small to modest portion of trait variability [5]. To address this, researchers have incorporated additional covariates in GWAS regression models to reduce residual variations to increase statistical power [5]. In addition, to distinguish total and direct effects of SNPs, conditional GWAS analysis has been proposed to adjust for some covariates [6]. In either case, a conditional SNP-trait association estimate is produced, which, however, may be biased [6, 7]: it is possible that both the SNP and unknown confounders may casually affect one of the covariates, making it a so-called collider; including a collider in analysis will induce conditional associations between SNPs and confounders that are otherwise truly independent. Furthermore, when the confounder is causal to both the trait and the covariate, this becomes problematic because it opens an indirect association pathway SNP → confounder → trait in addition to the direct association SNP → trait, biasing the conditional effect from its true value, inducing so-called collider bias. The magnitude of the bias depends on the association between the trait and covariates, as well as the SNP effect on the covariates [6].

Collider bias may result in spurious associations and false positives [8]. It will also have negative impact on downstream applications of GWAS results. For example, in Mendelian randomization (MR) analysis, biased SNP effect estimates can produce misleading causal estimates of the exposure on the outcomes [9].

Collider bias can also manifest in other scenarios, though not the focus of this paper. For instance, when analyzing disease progression using a case-only sample and conditioning on disease incidence, the presence of a shared confounder between disease incidence and progression can lead to spurious associations between SNPs associated with disease onset and that confounder [7, 10]. Consequently, the estimation of the direct SNP-on-progression effect will be biased by the indirect association path of SNP → confounder → disease prognosis [7].

Several approaches can be employed to correct collider bias [7, 11–13]. Under a simple model, a least squares approach reveals that the biased SNP-to-trait effect is equal to the true direct effect plus a bias term that is the product of the SNP-to-covariate effect and a slope [7]. Thus, estimating the bias term primarily involves approximating the slope. This estimation process resembles estimating the causal parameter of an exposure on an outcome using genetic variants as instrumental variables (IVs), for which various MR methods can be applied under certain IV assumptions [11]. Once a slope estimate is obtained, subtracting the bias term from the conditional effect estimate yields an (almost) unbiased estimate of the true direct effect. Previous studies have demonstrated that different MR approaches, such as inverse-variance weighted (IVW) regression and MR-Egger regression, can reduce type-I errors under certain conditions [11]. However, these approaches have limitations.

Firstly and most importantly, previous methods can only accommodate a single covariate and cannot effectively address collider bias induced by multiple covariates. In this paper, our primary objective is to mitigate bias resulting from the inclusion of multiple covariates in GWAS. Accordingly we propose application of multivariable Mendelian randomization (MVMR) methods to estimate the slope vector for multiple covariates being adjusted. Secondly, all MR methods impose more or less strong assumptions, especially on their requirements of valid IVs, to obtain a valid estimate of the slope vector [14]. Accordingly, we propose the use of multivariable MR constrained maximum likelihood (MVMR-cML), which is robust in the presence of invalid IVs as demonstrated before [14]. Consequently, we propose a bias correction approach utilizing MVMR-cML. In particular, here we prove the consistency and asymptotic normality of the collider bias-corrected estimator when MVMR-cML is applied, facilitating its valid use even in the presence of overlapping GWAS samples and when the SNP being tested and the SNPs being used as IVs in MVMR are in linkage disequilibrium (LD). Our proposed method with MVMR-cML can be applied to 1-sample, 2-sample and overlapping-sample settings with GWAS summary data. Nevertheless, for comparison, we also studied several other state-of-the-art MVMR methods.

The rest of the paper is structured as follows. Section 2 provides a comprehensive introduction to our bias-correction approach, outlining its key principles and methodology. In section 3, we proceed to assess the performance of the proposed methods with various MVMR across various scenarios through extensive simulations. The simulation results demonstrate that when a small number of covariates are adjusted, our approach effectively reduces collider bias while satisfactorily controlling type-I errors. However, as the number of covariates increases, the correction approach becomes less effective due to increased errors and uncertainties in parameter estimation. In section 4, we apply the methods to two UK Biobank GWAS datasets [5]. The first one considered a well-known example of a GWAS of waist-hip ratio (WHR) with adjustment for BMI as a single covariate, where our analysis indicated that collider bias had a minor impact on SNP-WHR association estimates. The second study utilized UK Biobank metabolomic data as multiple covariates to enhance statistical power of GWAS of BMI. Upon employing bias-correction, we observed that many previously significant SNPs were no longer significant with reduced effect size estimates, suggesting the likely presence of collider bias. More discussions are given in the final section.

## Description of the method

### Collider bias with multiple covariates

We consider a general problem of conditional association between a SNP (or any variable of interest) $G$ and an outcome $Y$, conditional on a vector of (quantitative) covariates $\mathbf{X}$. This type of conditional analysis is often performed when investigating the direct effect of $G$ on $Y$

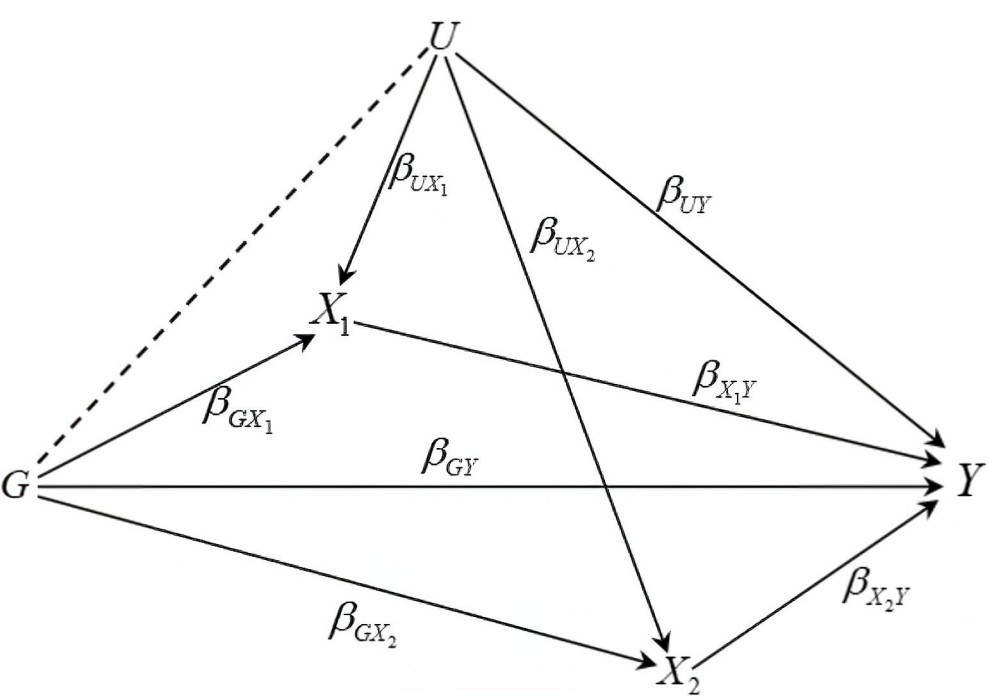

**Fig 1. Directed acyclic graph for outcome $Y$, SNP $G$, covariates $X_1$ and $X_2$, and confounder $U$.** Conditioning on colliders $X_1$ and $X_2$ induces a conditional association between $G$ and $U$, represented by the dashed line. This creates an indirect association between $G$ and $Y$ through the path $G \to U \to Y$, which will bias the estimation of the direct effect $\beta_{GY}$.

through pathways not mediated via $X$ [11], or when $X$ can explain a large proportion of the total variation of $Y$ to boost statistical power [5]. For the purpose of presentation, we first consider causal models, then at the end generalize the results to association analysis. We adopt the same terminology as used for collider bias in previous research [7, 11].

We first consider causal relationships among the variables in a simpler scenario with two covariates $X_1$ and $X_2$ as illustrated in Fig 1. Although we do not specify any causal relationship between covariates $X_1$ and $X_2$, they are allowed to be correlated. The total effect of $G$ on $Y$, denoted by $\beta_{GY}^{Total}$, consists of the direct effect $\beta_{GY}$ and the indirect effect mediated through $X$, represented by $\boldsymbol{\beta}_{GX}^T \boldsymbol{\beta}_{XY} = \beta_{GX_1}\beta_{X_1Y} + \beta_{GX_2}\beta_{X_2Y}$. To estimate the direct effect between $G$ and $Y$, one would condition on $X$. When there is no confounder present ($\beta_{UY} = 0$ and/or $\boldsymbol{\beta}_{UX} = (\beta_{UX_1}, \beta_{UX_2})^T = \boldsymbol{0}$), or when $X$ and $G$ are uncorrelated, conditioning on $X$ provides unbiased estimation of $\beta_{GY}$. However, when $X$ lies on the causal pathway from both $G$ and $U$, it becomes a collider. Consequently, conditioning on $X$ leads to an association between $G$ and $U$, resulting in an indirect association between $G$ and $Y$ through the path $G \to U \to Y$. This indirect association biases the estimation of $\beta_{GY}$ [7].

Throughout the paper, we assume that SNP $G$ and confounder $U$ are independent in the population [7], and the confounders of $G$ and $U$, such as population structure, are not present (or can be suitably adjusted). With GWAS summary statistics for $\beta_{GY}^{Total}$ and $\boldsymbol{\beta}_{GX}$, the direct effect $\beta_{GY}$ can be estimated using multitrait conditional/joint analysis (mtCOJO) [15], a previously proposed approach to estimate the SNP effects on $Y$ conditioning on multiple covariates $X_1, \cdots, X_p$. Specifically, the direct effect can be estimated as $\hat{\beta}_{GY} = \hat{\beta}_{GY}^{Total} - \hat{\boldsymbol{\beta}}_{GX}^T \boldsymbol{\beta}_{XY}$, where $\boldsymbol{\beta}_{XY}$

represents the effects of the covariates on the outcome when all covariates are jointly fitted, and can be transformed to $\boldsymbol{\beta}_{XY} = \boldsymbol{D}^{\frac{1}{2}} \boldsymbol{R}^{-1} \boldsymbol{D}^{\frac{1}{2}} \boldsymbol{d}_{XY}$ where $\boldsymbol{D}$ is a $p \times p$ diagonal matrix containing the SNP-based heritability of covariates, and $\boldsymbol{R}$ is a $p \times p$ matrix of genetic correlations of $X_1$, $\cdots$, $X_p$. Using GWAS summary data, $\boldsymbol{D}$ and $\boldsymbol{R}$ are estimated by linkage disequilibrium score (LDSC) regression [16, 17]. $\boldsymbol{d}_{XY}$ are the marginal effects of the covariates on $Y$ and can be estimated using MR. For more details about the mtCOJO approach, see Zhu et al. (2018) [15]. However, if only collider-biased conditional effect estimates are provided, mtCOJO cannot be employed to correct these effect estimates towards unbiasedness [11]. Consequently, other bias-correction approaches are necessary.

Previous studies have primarily focused on scenarios where only one covariate is included in the analysis, leading to the development of bias-correction approaches utilizing univariable Mendelian randomization (UVMR) [7, 11, 12]. In this paper, we consider more general situations where multiple covariates are incorporated into a GWAS regression, some or all of which may introduce collider bias. Our proposed bias-correction approach is applicable to GWAS summary data (for both the outcome and covariates). The problem is formulated as follows.

Suppose we have a SNP denoted as $G_i$, a vector of covariates represented as $\boldsymbol{X}$, an unmeasured confounder $U$, and a outcome/trait $Y$. The vector $\boldsymbol{X}$ consists of two components: $\boldsymbol{J} = (J_1, \cdots, J_{p_1})^T$ and $\boldsymbol{H} = (H_1, \cdots, H_{p_2})^T$. The variables in $\boldsymbol{J}$ are not associated with $G_i$ or $U$, but the elements in $\boldsymbol{H}$ may be influenced by both the SNP and the confounder, therefore inducing collider bias if included [6]. Fig 2 provides an illustration of the collider-bias problem in a multivariable scenario. When conditioning on the colliders $H_1$ and $H_2$, they induce an indirect association path $G_i \rightarrow U \rightarrow Y$, which can bias the estimation of the direct effect $\beta_{G_i Y}$. However, the covariates $J_1$ and $J_2$ are not associated with $G_i$ and $U$, and therefore do not introduce collider bias.

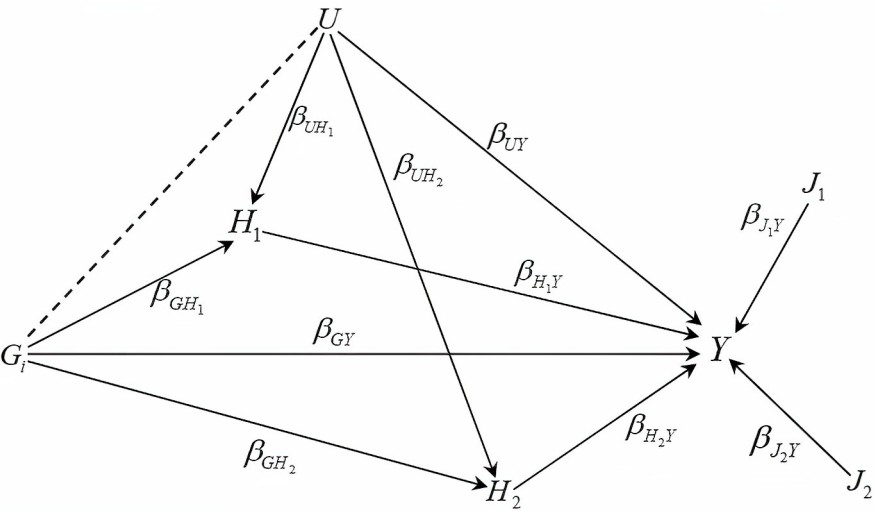

**Fig 2. Directed acyclic graph for outcome $Y$, SNP $G_i$, covariates $H$ and $J$, and confounder $U$.** Conditioning on the colliders $H_1$ and $H_2$ induces a conditional association between $G_i$ and $U$, represented by the dashed line. This creates an indirect association between $G_i$ and $Y$ through the path $G_i \rightarrow U \rightarrow Y$, which will bias the estimation of the direct effect $\beta_{G_i Y}$. $J_1$ and $J_2$ are not associated with $G_i$ and $U$, and therefore do not introduce bias.

We assume the following true causal model for SNP $G_i$, which extends previous studies to a more general multivariable scenario [7]:

$$H = \boldsymbol{\beta}_{G_iH}G_i + \mathbf{B}_{VH}V + \boldsymbol{\beta}_{UH}U + \boldsymbol{E}_{H}, \tag{1}$$

$$Y = \beta_{G_iY}G_i + \boldsymbol{\beta}_{XY}^{T}\boldsymbol{X} + \beta_{UY}U + E_{Y} \tag{2}$$

$$= \beta_{G_iY}G_i + \boldsymbol{\beta}_{JY}^{T}\boldsymbol{J} + \boldsymbol{\beta}_{HY}^{T}\boldsymbol{H} + \beta_{UY}U + E_{Y}. \tag{3}$$

In practice, one may include different covariates in the GWAS regression of $H$, we denote these covariates as $\boldsymbol{V} = (v_1, \cdots, v_q)^T$. To establish our analytical expression of the bias term, we need to assume $\boldsymbol{V}$ is uncorrelated with $U$ and the SNP, and hence the effect estimates $\hat{\boldsymbol{\beta}}_{G_iH}$ are not biased. Matrix $\boldsymbol{B}_{VH}$ consists of columns $\boldsymbol{\beta}_{v_sH}$ representing the effects of $v_s$ on $H$. Additionally, $\boldsymbol{E}_{H}$ and $E_Y$ represent the noises for $H$ and $Y$ respectively. For simplicity, we assume that $J$, $G_i$, $U$, $\boldsymbol{E}_{H}$, and $E_Y$ are pairwisely uncorrelated.

For each SNP $G_i$, the parameter of interest is its direct effect $\beta_{G_iY}$. However, since we do not observe the confounder $U$ in practice, we can only regress $Y$ on $G_i$ and $X$, and $U$ is absorbed into the error term $\epsilon$, which is thus correlated with $H$:

$$Y = \beta_{G_iY}^{C}G_i + \boldsymbol{\beta}_{JY}^{'T}\boldsymbol{J} + \boldsymbol{\beta}_{HY}^{'T}\boldsymbol{H} + \epsilon. \tag{4}$$

Consequently, the conditional effect $\beta_{G_iY}^{C}$ is biased for the true direct effect $\beta_{G_iY}$. In Section A in S1 Text, we derive that the bias term is the inner product of a vector $\boldsymbol{b}$ and $\boldsymbol{\beta}_{G_iH}$:

$$\beta_{G_iY}^{C} = \beta_{G_iY} + \boldsymbol{b}^{T}\boldsymbol{\beta}_{G_iH}, \tag{5}$$

$$\boldsymbol{b} = -\beta_{UY}Var(U)\mathbf{M}_{22}\boldsymbol{\beta}_{UH}. \tag{6}$$

$$\mathbf{M}_{22} = \left\{ \boldsymbol{\beta}_{UH}\boldsymbol{\beta}_{UH}^{T}Var(U) + Cov(\boldsymbol{E}_{H}) + \mathbf{B}_{VH}Cov(\boldsymbol{V})\mathbf{B}_{VH}^{T} \right. $$
$$\left. -\mathbf{B}_{VH}Cov(\boldsymbol{J}, \boldsymbol{V})^{T}Cov(\boldsymbol{J})^{-1}Cov(\boldsymbol{J}, \boldsymbol{V})\mathbf{B}_{VH}^{T} \right\}^{-1}. \tag{7}$$

This new result extends a previous one [7] for only univariate (i.e. one-dimensional) $H$ to that for multivariate (i.e. multi-dimensional) $H$. As previously [7], it holds under the assumption that the effects of the confounders $\boldsymbol{\beta}_{UH}$ and $\beta_{UY}$ remain constant across different SNPs, which is reasonable. In order to eliminate the bias term from $\beta_{G_iY}^{C}$, we need to estimate the slope vector $\boldsymbol{b}$, which can be done via MR.

In practice, when conducting GWAS marginally as usual for each SNP $G_i$, $G_i$ may not be causal to the outcome $Y$ (and $H$); instead, it may be simply in linkage disequilibrium with one or more causal SNPs. In this context, as in our real data examples, our above proposed analysis is to estimate the SNP's marginal and conditional associations, rather than its causal total and direct effects. But for simplicity we will still loosely use direct and total effects to refer to marginal and conditional associations in the sequel.

## Collider bias and MR

Eq (5) elucidates a relationship between collider bias correction and MVMR: both aim to estimate a linear relationship between instrumental effects on the outcome (conditioned on covariates) and instrumental effects on multiple covariates. The vector $\boldsymbol{b}$ in this context can be seen as the causal parameters in MVMR, while $\beta_{G_i Y}$ represents the pleiotropic effect of a SNP. Any MVMR approach can be employed to estimate $\boldsymbol{b}$, given that independent and valid IVs are available, although sample overlap should be carefully considered [18]. In MVMR, a valid IV must satisfy three assumptions:

A1: the IV is associated with at least one exposure conditional on the other exposures included in the model;

A2: the IV is independent of any confounder of each exposure-outcome pair;

A3: the IV is independent of the outcome conditional on all exposures included in the model and the confounders.

Under the model assumption in Eqs (1) and (2), as SNPs are assumed to be independent of the confounder $U$, valid instruments $G_i$'s satisfy the conditions $\boldsymbol{\beta}_{G_i \boldsymbol{H}} \neq \boldsymbol{0}$ (A1) and $\beta_{G_i Y} = 0$ (A3). In practice, GWAS summary statistics can be utilized to select instruments that affect the covariates. However, among the selected instruments, Condition A3 is likely to be violated since some SNPs are expected to have direct effects on $Y$. Hence, it is necessary to account for pleiotropy when estimating $\boldsymbol{b}$ in MVMR. In other words, although the direct effect $\beta_{G_i Y}$ is our target through out the paper, in the usual scenario of MVMR, it is pleiotropy effect that should be better avoided in analysis. This is a difference between collider bias correction and MVMR estimation [11].

Previous studies mainly focused on the univariable case, where the GWAS of $Y$ includes only one covariate and $\boldsymbol{b}$ reduces to a scalar $b$. Weighted regression, incorporating an intercept to accommodate for pleiotropy, was employed to obtain $\hat{b}$ [7, 11]. This approach is analogous to univariable MR-Egger (UV-Egger) regression and can be extended to the multivariable case. However, this approach assumes independence between instrumental effects on covariates and the direct effects $\beta_{G_i Y}$ (referred to as the InSIDE assumption) [19]. When there are correlated pleiotropic effects, this assumption is violated, leading to biased estimation of $\boldsymbol{b}$ [19]. In practice, the InSIDE assumption may be violated when the outcome and covariates share common biological mechanisms [11]. Other robust MVMR methods, such as MVMR-Lasso and MVMR-median, can be used to estimate $\boldsymbol{b}$ [20]. A previous numerical study shows that, compared to multvariable MR-Egger (MVMR-Egger) regression, these methods produce a more precise estimate under correlated pleiotropy [14, 20]. However, they may still be biased when too many invalid IVs are used for analysis [14]. Another bias correction method, known as Slope-Hunter has been developed to account for correlated pleiotropy [12]. Numerical examples demonstrate its efficacy in reducing collider bias unless there is a strong negative correlation between pleiotropic effects [12]. However, it (along with aforementioned and many other robust MR methods) is limited to the univariable case and has not yet been generalized to situations with multiple covariates (or exposures).

It is important to note other drawbacks of MR-Egger regression. In the univariate case, the performance of Egger regression depends on the orientation of SNPs [21]. To ensure the independence of analysis from the reported reference alleles, SNPs were flipped to have positive effects on the risk factor [19], known as default coding. However, violations of the InSIDE assumption can occur when some, but not all, SNPs are re-oriented to achieve positive associations with the exposure [21]. And this problem may also arise in a multivariate scenario. The

previous literature also indicated that under unbalanced pleiotropy, default coding may severely bias the causal estimate of Egger regression. [21]. In our simulation, where only balanced pleiotropy was present, Egger regression was nearly unbiased in the univariate case, but the variance of causal estimate was large. This is consistent with a previous study [21]. Furthermore, the implementation of UV-Egger regression requires no measurement error in the SNP-exposure association (known as the NOME assumption). Hence, the SNP effects on exposures are assumed to be known without accounting for their uncertainty. When the InSIDE assumption holds, the presence of uncertainty in the SNP effects on exposures can bias the estimate towards 0 [22]. In the context of collider bias, where the SNP effects on covariates are obtained from GWAS summary data, this can lead to underestimation of the magnitude of bias and subsequently result in under-adjustment of the conditional estimates $\hat{\beta}^C_{G_i Y}$ [11]. This issue has not been extensively studied for MVMR-Egger regression [19]. In Table O in S1 Text, some simulation results indicate that, when the InSIDE assumption was violated, MVMR-Egger yielded biased estimates $\hat{\boldsymbol{b}}$.

The previous discussion highlights the limitations of existing approaches. Therefore, it is crucial to employ a robust MVMR method that can accurately estimate the slope vector $\boldsymbol{b}$ and thus correct for collider bias. MVMR-cML emerges as a strong competitor as it remains robust even in the presence of invalid IVs that violate all three IV assumptions [14]. Previous simulations have demonstrated the competitive performance of MVMR-cML among robust MVMR approaches in the presence of correlated and uncorrelated pleiotropy [14]. Hence, in our proposal, we estimate $\boldsymbol{b}$ using MVMR-cML, and obtain the estimate of $\beta_{G_i Y}$ by subtracting the bias from the conditional estimate $\hat{\beta}^C_{G_i Y}$. We refer this method as MVMR-cML-bias-correction. More details are given below.

## Bias correction via MVMR-cML

Given the GWAS summary datasets $\{\hat{\beta}_{G_i H_1}, \cdots, \hat{\beta}_{G_i H_{p_2}}, \hat{\beta}^C_{G_i Y}, \hat{\sigma}_{G_i H_1}, \cdots, \hat{\sigma}_{G_i H_{p_2}}, \hat{\sigma}^C_{G_i Y}\}^m_{i=1}$ of covariates $H_1, \cdots, H_{p_2}$ and the outcome $Y$, computed by regressions in Eqs (1) and (4). We first select (approximately) independent IVs $Z_j$ through a pruning process. Subsequently, we identify the corresponding instrumental effect estimates $\{\hat{\beta}_{Z_j H_1}, \cdots, \hat{\beta}_{Z_j H_{p_2}}, \hat{\beta}^C_{Z_j Y}, \hat{\sigma}_{Z_j H_1}, \cdots, \hat{\sigma}_{Z_j H_{p_2}}, \hat{\sigma}^C_{Z_j Y}\}^l_{j=1}$ based on the significant marginal associations between each $Z_j$ and at least one of the covariates in $\boldsymbol{H}$. Let $\mathcal{V}^* = \{Z^*_1, \cdots, Z^*_{l^0}\}$ represent the true (unknown) set of valid IVs, where $|\mathcal{V}^*| = l^0$. Through out this paper we use a superscript 0 to denote true values of parameters.

As previously [14], given the usual large sample size of GWAS, it is reasonable to assume a multivariate normal model for SNP effect estimates:

$$\begin{pmatrix} \hat{\beta}^C_{Z_j Y} \\ \hat{\boldsymbol{\beta}}_{Z_j \boldsymbol{H}} \end{pmatrix} \sim N \left\{ \begin{pmatrix} (\boldsymbol{b}^0)^T \boldsymbol{\beta}^0_{Z_j \boldsymbol{H}} + \beta^0_{Z_j Y} \\ \boldsymbol{\beta}^0_{Z_j \boldsymbol{H}} \end{pmatrix}, \boldsymbol{\Sigma}_j \right\}, \quad j = 1, \cdots, l. \tag{8}$$

The non-diagonal elements of covariance matrix $\boldsymbol{\Sigma}_j$ capture the correlations among the summary statistics for the outcome and covariates due to overlapping samples.

The correlation parameters in $\boldsymbol{\Sigma}_j$ can be estimated using either null z-scores in GWAS summary data [23] or LDSC regression. [24] Throughout this paper, we approximate $\boldsymbol{\Sigma}_j$ using null z-scores. In the context of MVMR-cML, we assume that $\boldsymbol{\Sigma}_j$ is either known or well-estimated using GWAS summary data [14].

The log-likelihood function of $\boldsymbol{b}, \boldsymbol{\beta}_{Z_j H}$ and $\beta_{Z_j Y}$ is:

$$L\left(\boldsymbol{b}, \boldsymbol{\beta}_{Z_j H}, \beta_{Z_j Y} \middle| \hat{\boldsymbol{\beta}}_j, \boldsymbol{\Sigma}_j\right) = -\frac{1}{2}\sum_{j=1}^{l}\left(\hat{\boldsymbol{\beta}}_j - \boldsymbol{\beta}_j\right)^T \boldsymbol{\Sigma}_j^{-1}\left(\hat{\boldsymbol{\beta}}_j - \boldsymbol{\beta}_j\right),\tag{9}$$

where $\boldsymbol{\beta}_j = (\boldsymbol{b}^T\boldsymbol{\beta}_{Z_j H} + \beta_{Z_j Y}, \beta_{Z_j H_1}, \cdots, \beta_{Z_j H_{p_2}})^T$ and $\hat{\boldsymbol{\beta}}_j = (\hat{\beta}_{Z_j Y}^C, \hat{\beta}_{Z_j H_1}, \cdots, \hat{\beta}_{Z_j H_{p_2}})^T$. Under the constraint that the number of invalid IVs is $K$, we estimate the unknown parameters $\boldsymbol{b}, \boldsymbol{\beta}_{Z_j H}$ and $\beta_{Z_j Y}$ by solving the following constrained maximum likelihood problem:

$$\{\hat{\boldsymbol{b}}, \boldsymbol{\beta}_{Z_j H}, \beta_{Z_j Y}\} = \quad \operatorname{argmax}_{\{\boldsymbol{b}, \boldsymbol{\beta}_{Z_j H}, \beta_{Z_j Y}\}} L(\boldsymbol{b}, \boldsymbol{\beta}_{Z_j H}, \beta_{Z_j Y} | \hat{\boldsymbol{\beta}}_j, \boldsymbol{\Sigma}_j)\tag{10}$$

$$\text{subject to } \sum_{j=1}^{l} I(\beta_{Z_j Y} \neq 0) = K.\tag{11}$$

For a given number of invalid IVs, $K$, a coordinate descent-like algorithm is implemented to obtain the estimates $\hat{\boldsymbol{b}}(K)$ and $\{\tilde{\boldsymbol{\beta}}_{Z_j H}(K), \tilde{\beta}_{Z_j Y}(K)\}_{j=1}^{l}$. The selection of $K$ is performed using the Bayesian information criterion ($BIC$) from a candidate set $\mathcal{K} = \{0, 1, \cdots, l - p_2 - 1\}$:

$$BIC(K) = -2L(\hat{\boldsymbol{b}}(K), \tilde{\boldsymbol{\beta}}_{Z_j H}(K), \tilde{\beta}_{Z_j Y}(K) | \hat{\boldsymbol{\beta}}_j, \boldsymbol{\Sigma}_j) + K \log(N).$$

where $N$ is the minimum sample size of all GWAS datasets used for analysis. The value of $K$ ranges from 0 to $l - p_2 - 1$, taking into account the multivariable plurality condition [14]. When $K = 0$, it indicates that all IVs are valid. By minimizing $BIC$, we determine the estimate $\hat{K}$, and final $\hat{\boldsymbol{b}} = \hat{\boldsymbol{b}}(\hat{K})$ and $\hat{\mathcal{V}}^* = \{Z_j | \hat{\beta}_{Z_j Y}(\hat{K}) = 0\}$. A consistently estimated covariance matrix of $\hat{\boldsymbol{b}}$, denoted as $\hat{\boldsymbol{\Sigma}}_{\hat{\boldsymbol{b}}}$, can be obtained by the observed Fisher information matrix from the likelihood using all selected valid IVs in $\hat{\mathcal{V}}^*$. This approach is referred to as MVMR-cML-BIC, following the previous terminology [14]. Under mild conditions, MVMR-cML consistently select the set of valid IVs. Specifically, as $N \to +\infty$, we have $P(\hat{\mathcal{V}}^* = \mathcal{V}^*) \to 1$. Additionally, the distribution of the standardized difference $\boldsymbol{\Sigma}_{\hat{\boldsymbol{b}}}^{-\frac{1}{2}}(\hat{\boldsymbol{b}} - \boldsymbol{b}^0)$ approaches a multivariable standard normal distribution [14]. $\boldsymbol{\Sigma}_{\hat{\boldsymbol{b}}}$ is substituted by $\hat{\boldsymbol{\Sigma}}_{\hat{\boldsymbol{b}}}$ in practice. For better finite-sample performance, a data perturbation approach was proposed to account for statistical uncertainty in model selection [14]. However, in the current paper, this approach is not utilized due to its time-consuming nature.

After obtaining the estimated slope vector $\hat{\boldsymbol{b}}$, we can calculate the bias-corrected estimate as

$$\hat{\beta}_{G_i Y} = \hat{\beta}_{G_i Y}^C - \hat{\boldsymbol{b}}^T \hat{\boldsymbol{\beta}}_{G_i H}.\tag{12}$$

We have the following result for the desired statistical property of $\hat{\beta}_{G_i Y}$.

**Theorem 1** *Under some mild conditions, as the sample size $N \to +\infty$, the bias-corrected estimator $\hat{\beta}_{G_i Y} = \hat{\beta}_{G_i Y}^C - \hat{\boldsymbol{b}}^T \hat{\boldsymbol{\beta}}_{G_i H}$ is consistent and has an asymptotic normal distribution:*

$$(\hat{\beta}_{G_i Y} - \beta_{G_i Y}^0)/\sigma_{G_i Y} \xrightarrow{D} N(0, 1),\tag{13}$$

*where $\beta_{G_i Y}^0$ is the true direct effect of $G_i$ on $Y$, and $\sigma_{G_i Y}^2$ is the variance of $\hat{\beta}_{G_i Y}$.*

A consistent estimator $\hat{\sigma}^2_{G_iY}$ of $\sigma^2_{G_iY}$ is shown in Section B.2 in S1 Text. For MVMR-cML, the variance estimator $\hat{\sigma}^2_{G_iY}$ properly accounts for correlations among the GWAS summary data, as well as LD between the target SNP $G_i$ and IVs. More details, including the proof and conditions of Theorem 1 and the analytical expression of $\hat{\sigma}^2_{G_iY}$, are given in the Section B in S1 Text.

Denote the elements of $\hat{\boldsymbol{b}}$ as $\hat{b}_k$, and the diagonal elements of $\hat{\boldsymbol{\Sigma}}\hat{b}$ as $\hat{\sigma}^2_{\hat{b}_k}$. Assuming that the estimated slope vector $\hat{\boldsymbol{b}}$, $\hat{\beta}^C_{G_iY}$, and the elements in $\hat{\boldsymbol{\beta}}_{G_iH}$ are mutually independent, we have the simplified expression of $\hat{\sigma}^2_{G_iY}$ as follows [7, 12]:

$$\hat{\sigma}^2_{G_iY} = \hat{\boldsymbol{\beta}}^T_{G_iH}\hat{\boldsymbol{\Sigma}}_{\hat{b}}\hat{\boldsymbol{\beta}}_{G_iH} + \sum_{k=1}^{p_2}\hat{b}_k^2\hat{\sigma}^2_{G_iH_k} + \sum_{k=1}^{p_2}\hat{\sigma}^2_{\hat{b}_k}\hat{\sigma}^2_{G_iH_k} + \hat{\sigma}^{2C}_{G_iY}. \tag{14}$$

Although the independence assumption does not hold in practice, in our simulation and real data applications, Eq (14) often yielded similar variance estimates. In general, the variance increases as more covariates are adjusted. Hence, as expected, adjusting for too many heritable covariates leads to a sacrifice of power.

Note that our method is applicable to GWAS summary data for both the heritable covariates and the trait of interest.

## Other methods

Note that our proposed bias-correction method can be applied with various MVMR methods other than MVMRcML, such as MVMR-Egger, MVMR-IVW, MVMR-Lasso, and MVMR-median [20]; we compared the performance of these methods in simulations and real data analyses. For other MVMR methods to be applicable here as well as possible, we assumed the asymptotic normality of their estimates and applied (14) to estimate the variance of their SNP effect estimate after bias correction, where the covariance matrix of $\boldsymbol{b}$ was estimated by data perturbation [14]. Note that it is unknown how to estimate the correlations between $\hat{\boldsymbol{b}}$ and $\hat{\boldsymbol{\beta}}_{G_iH}$ or $\hat{\beta}^C_{G_iY}$ for other MVMR methods, while they can be estimated for MVMR-cML. The corresponding bias-correction procedure was named after the MVMR method, for example, as MVMR Egger bias correction. In MVMR-Egger regression, we reoriented the SNPs such that they all had positive effects on the first covariate $H_1$ [19].

In addition, in cases with only one heritable covariate (i.e. $p_2 = 1$), we also applied two bias-correction approaches of Dudbridge et al. (2019) [7] and Mahmoud et al (2022) [12]; following the previous literature [12], we denote these two methods respectively as **DHO** and **Slope-Hunter (SH)**.

## Simulation setups

We conducted a simulation study to assess the performance of different MVMR methods in the one-sample setting (i.e. where all GWAS data were based on the same sample of individuals) under two scenarios: one involving independent SNPs with no pleiotropy, and the other using (weakly) correlated SNPs with pleiotropy. In the former, independent SNPs were randomly generated; in the latter, we first pruned the UK Biobank (UKB) genotype data with correlation coefficients 0.1, then we randomly sampled 1000 SNPs from chromosome 1. Among the 1000 SNPs, 30 independent SNPs were randomly selected as IVs; 50 SNPs affected $\boldsymbol{H}$ only, 50 affected $Y$ only, 50 affected both $\boldsymbol{H}$ and $Y$; the remaining ones were null SNPs having no effect on either the covariates or the outcome. Before drawing the SNPs, we partitioned chromosome 1 into 133 independent blocks [25]. The 150 non-null SNPs were drawn from the first 10 blocks, and the 850 null SNPs were drawn from the last 10 blocks. Hence, the non-null

SNPs were independent of the null SNPs. We also drew 3000 independent null SNPs from other chromosomes, whose z-scores were used to estimate the correlations of summary statistics [24]. These 3000 null SNPs were not presented in simulation results. When estimating the correlation of summary statistics, we used a p-value 0.1 to select null z-cores for the covariates or the outcome, and approximately 2000 null z-scores were selected for each pair of summary data. We utilized correlated SNPs in order to compare the variance estimator in Eq (14) with our newly proposed variance estimator of MVMR-cML, the later of which can account for LD among SNPs. However, in our simulation, the correlations among SNPs had minor influence on variance estimation. All genotypes $G_i$ were centered to have a sample mean of 0. In order to simulate correlated pleiotropy, for the 50 SNPs affecting both $\boldsymbol{H} = (H_1, \cdots, H_{p_2})^T$ and $Y$, their effects on $Y$ ($\beta_{G_i Y}$) and the first covariate $H_1$ ($\beta_{G_i H_1}$) were generated from a bivariate normal distribution with mean 0, variance 1, and a constant correlation $\rho$ being 0, 0.5 or −0.5. All other SNP effects, as well as the causal effects from the covariates to the outcome ($\boldsymbol{\beta_{HY}}$) were independently drawn from a standard normal distribution. All SNP effects were predetermined before initiating the simulation. The confounder $U$ and error terms $E_{\boldsymbol{H}}$, $E_Y$ were drawn from normal distributions with a mean of 0. $U$ accounted for 40% of the unknown variance in both $Y$ and the covariates within $\boldsymbol{H}$. Each error term contributed 10% of the total variance in the covariates or outcome. The values of $\boldsymbol{H}$ and $Y$ were subsequently determined based on the equations below:

$$\boldsymbol{H} = \sum_{i=1}^{4000} \boldsymbol{\beta}_{G_i \boldsymbol{H}} G_i + \boldsymbol{\beta}_{U\boldsymbol{H}} U + E_{\boldsymbol{H}}, \tag{15}$$

$$Y = \sum_{i=1}^{4000} \beta_{G_i Y} G_i + \beta_{UY} U + \boldsymbol{\beta}_{\boldsymbol{HY}}^T \boldsymbol{H} + E_Y. \tag{16}$$

We conducted the simulation under two scenarios with 30% and 50% invalid IVs respectively. The invalid IVs were taken from those having pleiotropic effects, while the valid IVs were taken from those affecting the covariates only. The GWAS of $Y$ was performed using $\boldsymbol{H}$ as covariates, while the GWAS of each element in $\boldsymbol{H}$ only included the SNPs. The simulation set-ups closely followed those in a previous study [7]. We mainly presented the results for the SNPs suffering from collider bias, i.e., those SNPs affecting the covariates.

## Verification and comparison

In each simulation scenario, we varied the dimension of $\boldsymbol{H} = (H_1, \cdots, H_{p_2})$, denoted as $p_2$, considering dimensions of 1, 2 and 4. Here we only present the main simulation with correlated SNPs and uncorrelated pleiotropy (30% invalid IVs) while additional details regarding other scenarios can be found in the Section F in S1 Text.

   We applied our bias-correction method to the simulated GWAS summary data, employing a significance level of 0.05 to calculate type-I error rates and power. To gauge the impact of collider bias and assess the effectiveness of the bias-correction methods, we computed the probability of type-I error. Specifically, the probability of rejecting the null hypothesis $H_0 : \beta_{G_i Y} = 0$ for each SNP that exerted no direct effect on $Y$, respectively before and after bias correction. Recognizing that different SNPs might exhibit distinct type-I error rates, we calculated the average across all these SNPs, providing an overall measure. In evaluating the power of detecting the effect $\beta_{G_i Y}$, we computed the empirical probability of rejecting the null hypothesis. Power calculations were performed individually for each SNP affecting $Y$, and the results were averaged over different SNPs to derive an overall measure. Given that our bias-correction

approach may impact power, we specifically identified SNPs exhibiting the largest increase or decrease in power after applying MVMR-cML-bias-correction. This allowed us to scrutinize the bias-correction methods' impact on power in extreme cases.

To highlight the bias-correction results for a single SNP, we randomly selected two SNPs with collider bias (i.e., SNPs affecting covariates) and provided their corresponding effect estimates, type-I error rates and power in the Tables P-R and Tables W-Y in S1 Text.

## Dramatically inflated Type-I errors were better controlled after bias correction

In Table 1, we provide a comprehensive overview of empirical type-I error rates and power across different SNPs, along with the sample standard deviations (SD) of the point estimates. Overall, when $p_2 = 1$, all MVMR methods demonstrated effectiveness in mitigating inflated type-I errors resulting from collider bias. Notably, for SNPs affecting only the covariates, type-I error rates were close to the nominal level after bias correction of each MVMR method. For example, when $p_2 = 1$, the empirical type-I error rate of SNPs affecting $H$ but not $Y$ decreased from 0.79 to 0.05 after applying MVMR-cML for bias correction. As pointed out by one reviewer, the method of Dudbridge et al (2019) [7] is also robust to overlapping samples as confirmed here by the good performance of DHO.

When $p_2 \geq 2$, all MVMR methods were less effective: all the type-I error rates remained slightly above 0.05 after correction, but MVMR-cML and MVMR-median gave smaller type-I error rates than other methods. Two reasons contributed to the minor inflation of type-I errors. Firstly, for SNPs solely influencing the covariates, their summary statistics $\hat{\boldsymbol{\beta}}_{G_iH}$, exhibited slight biases owing to inter-SNP correlations. When $p_2 = 1$, this bias was minimal and thus had negligible impact on bias correction. However, with the inclusion of more covariates in the analysis, the collective estimation errors of the entire vector $\hat{\boldsymbol{\beta}}_{G_iH}$ increased. This problem also exist in the estimation of $\boldsymbol{b}$. As shown in Table 2, $\hat{\boldsymbol{b}}$ were slightly biased from the true values due to invalid IVs. Even the deviation is small for each element, the aggregated estimation error of $\hat{\boldsymbol{b}}$ increased and became influential as more covariates joined the analysis. According to Eq (12), this would reduce the accuracy of our bias-correction method. As for null SNPs, although their summary statistics remained unbiased, integrating additional covariates in GWAS augmented the uncertainty in estimating $\boldsymbol{\beta}_{G_iH}$, making it more challenging for bias-correction to approximate the true direct effect $\beta_{G_iY}$. Consequently, the effect estimates of a few null SNPs might deviate from 0 after bias correction, thereby slightly inflating type-I errors.

## Collider bias was mitigated after bias correction

Fig 3 illustrates the mean estimates of $\beta_{G_iY}$ (averaged over 1000 simulations) both before and after bias correction against the true (direct or conditional) effects. Here we exclusively present figures for MVMR-cML, representing our proposed method. The complete set of results is available in the Section F in S1 Text. Across various scenarios, all MVMR methods produced similar figures. Before correction, numerous points deviated from the identity line, indicating the presence of collider bias in the conditional effects $\beta_{G_iY}^C$ for these SNPs. Fig 3 demonstrates the effectiveness of MVMR-cML in eliminating or reducing bias, particularly when $p_2 = 1$. This is evident as most points aligned closely around the identity line after bias correction. However, when $p_2 > 1$, our proposed method might not entirely eliminate bias, and many points slightly deviated from the identity line. This deviation was attributed to the introduction

**Table 1. Empirical type-I error rate (for SNPs underlined) and power with and without bias correction in the presence of 30% invalid IVs.** Sample standard deviations (SD) are given in parenthesis.

| Dimension of H | 1 | | | | | | | | 2 | | | | | | 4 | | | | | |
| --- | --- | --- | --- | --- | --- | --- | --- | --- | --- | --- | --- | --- | --- | --- | --- | --- | --- | --- | --- | --- |
| Bias correction | No | cML | Egger | IVW | Lasso | Median | DHO | SH | No | cML | Egger | IVW | Lasso | Median | No | cML | Egger | IVW | Lasso | Median |
| Null SNPs | 0.05 | 0.05 | 0.05 | 0.05 | 0.05 | 0.05 | 0.05 | 0.05 | 0.05 | 0.06 | 0.06 | 0.06 | 0.06 | 0.06 | 0.05 | 0.06 | 0.10 | 0.10 | 0.10 | 0.10 |
| (SD) | (0.01) | (0.01) | (0.01) | (0.01) | (0.01) | (0.01) | (0.01) | (0.01) | (0.01) | (0.01) | (0.01) | (0.01) | (0.01) | (0.01) | (0.01) | (0.01) | (0.01) | (0.01) | (0.01) | (0.01) |
| All SNPs affecting H but not Y | 0.73 | 0.05 | 0.05 | 0.05 | 0.05 | 0.05 | 0.07 | 0.05 | 0.74 | 0.06 | 0.07 | 0.09 | 0.09 | 0.06 | 0.71 | 0.08 | 0.17 | 0.12 | 0.12 | 0.09 |
| (SD) | (0.04) | (0.03) | (0.03) | (0.03) | (0.03) | (0.03) | (0.05) | (0.03) | (0.03) | (0.04) | (0.04) | (0.04) | (0.04) | (0.04) | (0.04) | (0.04) | (0.07) | (0.04) | (0.04) | (0.04) |
| All SNPs affecting Y only | 0.34 | 0.26 | 0.26 | 0.26 | 0.26 | 0.26 | 0.26 | 0.26 | 0.65 | 0.54 | 0.56 | 0.55 | 0.55 | 0.56 | 0.77 | 0.70 | 0.72 | 0.73 | 0.73 | 0.73 |
| (SD) | (0.05) | (0.05) | (0.05) | (0.05) | (0.05) | (0.05) | (0.05) | (0.05) | (0.04) | (0.05) | (0.05) | (0.05) | (0.05) | (0.05) | (0.03) | (0.04) | (0.04) | (0.04) | (0.04) | (0.04) |
| All SNPs affecting both H and Y | 0.76 | 0.26 | 0.24 | 0.26 | 0.26 | 0.25 | 0.26 | 0.25 | 0.69 | 0.52 | 0.54 | 0.53 | 0.53 | 0.52 | 0.82 | 0.66 | 0.63 | 0.66 | 0.66 | 0.62 |
| (SD) | (0.04) | (0.05) | (0.04) | (0.05) | (0.05) | (0.05) | (0.05) | (0.05) | (0.04) | (0.05) | (0.05) | (0.05) | (0.05) | (0.05) | (0.03) | (0.05) | (0.04) | (0.04) | (0.04) | (0.04) |
| SNP with greatest increase in power | 0.23 | 0.87 | 0.87 | 0.87 | 0.87 | 0.87 | 0.87 | 0.87 | 0.05 | 0.98 | 0.99 | 0.98 | 0.98 | 0.98 | 0.28 | 0.95 | 0.99 | 0.98 | 0.98 | 0.97 |
| (SD) | (0.42) | (0.34) | (0.34) | (0.34) | (0.34) | (0.34) | (0.33) | (0.34) | (0.22) | (0.13) | (0.09) | (0.13) | (0.13) | (0.14) | (0.45) | (0.21) | (0.11) | (0.13) | (0.13) | (0.16) |
| SNP with greatest decrease in power | 1.00 | 0.05 | 0.01 | 0.03 | 0.03 | 0.02 | 0.03 | 0.05 | 1.00 | 0.05 | 0.04 | 0.06 | 0.06 | 0.04 | 1.00 | 0.06 | 0.07 | 0.09 | 0.09 | 0.03 |
| (SD) | (0.00) | (0.22) | (0.10) | (0.18) | (0.18) | (0.13) | (0.18) | (0.21) | (0.00) | (0.23) | (0.19) | (0.24) | (0.24) | (0.20) | (0.00) | (0.24) | (0.26) | (0.24) | (0.28) | (0.18) |

**Table 2. True values and mean estimates of $b$ with $\rho = 0$ and 30% invalid IVs.** Sample standard deviations (SD), mean standard errors (Mean SE) are given in parenthesis.

| Dimension of $b$ | 1 | 2 | | 4 | | | |
|---|---|---|---|---|---|---|---|
| $b$ | $b_1$ | $b_1$ | $b_2$ | $b_1$ | $b_2$ | $b_3$ | $b_4$ |
| True value | −3.26 | −1.19 | −1.22 | −0.48 | −0.64 | −0.46 | −0.39 |
| MVMR-cML | −3.36 | −1.17 | −1.26 | −0.51 | −0.83 | −0.41 | −0.37 |
| (SD) | (0.10) | (0.07) | (0.07) | (0.06) | (0.06) | (0.04) | (0.07) |
| (Mean SE) | (0.09) | (0.05) | (0.06) | (0.04) | (0.05) | (0.04) | (0.03) |
| MVMR-Egger | −3.41 | −1.05 | −1.25 | −0.69 | −0.74 | −0.36 | −0.42 |
| (SD) | (0.16) | (0.10) | (0.05) | (0.07) | (0.05) | (0.04) | (0.03) |
| (Mean SE) | (0.21) | (0.11) | (0.06) | (0.08) | (0.05) | (0.05) | (0.03) |
| MVMR-IVW | −3.32 | −1.26 | −1.24 | −0.56 | −0.72 | −0.41 | −0.42 |
| (SD) | (0.09) | (0.05) | (0.05) | (0.04) | (0.05) | (0.03) | (0.03) |
| (Mean SE) | (0.09) | (0.05) | (0.06) | (0.03) | (0.05) | (0.04) | (0.03) |
| MVMR-Lasso | −3.32 | −1.26 | −1.24 | −0.56 | −0.72 | −0.41 | −0.42 |
| (SD) | (0.09) | (0.05) | (0.05) | (0.04) | (0.05) | (0.03) | (0.03) |
| (Mean SE) | (0.09) | (0.05) | (0.06) | (0.03) | (0.05) | (0.04) | (0.03) |
| MVMR-Median | −3.36 | −1.21 | −1.22 | −0.56 | −0.73 | −0.42 | −0.43 |
| (SD) | (0.10) | (0.06) | (0.06) | (0.04) | (0.05) | (0.04) | (0.04) |
| (Mean SE) | (0.13) | (0.08) | (0.09) | (0.06) | (0.08) | (0.05) | (0.06) |
| DHO | −3.54 | NA | NA | NA | NA | NA | NA |
| (SD) | (0.23) | | | | | | |
| (Mean SE) | (0.17) | | | | | | |
| SH | −3.33 | NA | NA | NA | NA | NA | NA |
| (SD) | (0.11) | | | | | | |
| (Mean SE) | (0.12) | | | | | | |

of more covariates in $H$. As mentioned before, this increased the collective estimation error in the vector $\hat{\boldsymbol{\beta}}_{G_iH}$ and reduced the accuracy of $\hat{\beta}_{G_iY}$. Additionally, invalid IVs might introduce slight bias into the estimate of $b$ as shown in Table 2. Even the bias for each element of $\hat{\boldsymbol{b}}$ was small, as more covariates were included in analysis, the aggregated estimation error of the entire vector $\hat{\boldsymbol{b}}$ got large. Consequently, achieving precise estimation became challenging, resulting in a slight bias even after correction. Furthermore, our bias correction approach, as described in Eq (14), increased the variance of the corrected effect estimate. This was evident in the figures, where the vertical bars representing mean standard errors (SEs) became longer after correction. Hence there is a trade-off between type I error control and statistical power, as larger SEs reduce the power to detect true associations between SNPs and the outcome $Y$. Therefore, while our correction approach effectively addressed collider bias, it came at the expense of sacrificing some power, especially in scenarios with more covariates in $H$.

In S1 Text, Tables P-R present a summary of mean effect estimates, empirical type-I error rates and power for randomly selected SNPs exhibiting collider bias. Sample standard deviation (SD) and mean standard error (Mean SE) are provided in parentheses. Without bias correction, the empirical type-I error rates were dramatically inflated due to collider bias. All MVMR methods effectively addressed the bias, thus reducing type-I errors. For any single SNP, its estimation accuracy was improved, and the effect estimates were closer to the true value after bias correction. For example, in Table Q in S1 Text, when $\rho = 0$, the effect estimate of the first SNP, whose true effect was 0, changed from −3.54 to 0.09 after MVMR-cML-bias-

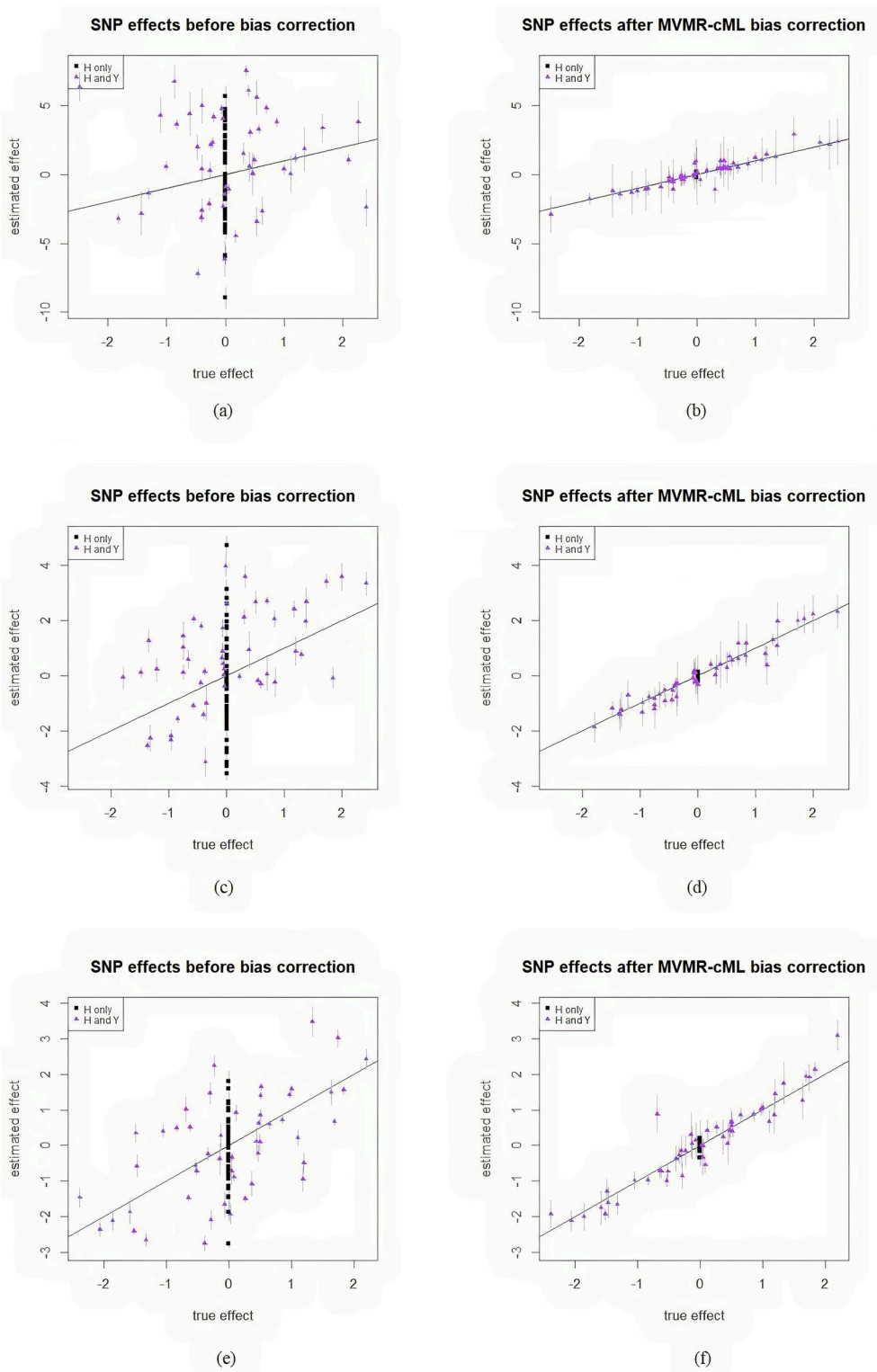

**Fig 3. Mean estimates of the effects of SNPs having collider bias with $\rho$ = 0 and 30% invalid IVs.** Horizontal coordinates are for the true effects, vertical coordinates are for the estimated effects. Vertical bars are the means of standard errors averaged over 1000 repetitions. In the legends, "**H** only" means the SNPs affecting only the covariates; "**H** and *Y*" means the SNPs affecting both the covariates and outcome. (a)-(b): $p_2$ = 1; (c)-(d): $p_2$ = 2; (e)-(f): $p_2$ = 4.

correction, but with the standard error increased from 0.22 to 0.34. Correspondingly, the type-I error rate decreased from 1 to 0.02, other MVMR methods produced similar results. For the SNPs affecting both the covariates and outcome, most MVMR methods provided a more accurate effect estimate after bias correction. For example, in Table R in S1 Text, when $\rho$ = 0.5, the effect estimate of the third SNP (whose true effect was −0.78) was −0.3 before correction; after MVMR-Lasso-bias-correction, it was −0.55.

## Estimation challenges with more covariates

Table 2 provides the mean estimates of $\boldsymbol{b}$ for each MVMR method. As mentioned earlier, all MVMR methods yielded relatively accurate estimates of $\boldsymbol{b}$. However, as mentioned before, when $p_2$ = 4, even the the estimation bias for each element is small, the collective estimation error of the entire vector $\hat{\boldsymbol{b}}$, as well as $\hat{\boldsymbol{\beta}}_{G_iH}$ might be large. Consequently, the effect estimates of some SNPs was inaccurate after bias correction. For example, in Table R in S1 Text, when $\rho$ = 0, the mean effect estimate of the forth SNP (whose true value was 0.65) was 0.59 before correction, and became 1.09 after MVMR-Egger-bias-correction. When $p_2$ = 1, Egger regression had a larger standard error of $\hat{\boldsymbol{b}}$ than other methods. This is consistent with the previous literature: Egger regression with the default coding yielded a larger variance for the causal estimate, compared to other methods, such as IVW regression [21].

## Other simulation results

In Section F in S1 Text, additional simulation results are presented. When the analysis involved a higher proportion of invalid IVs (50%), or when the InSIDE assumption was violated, certain MVMR methods, such as MVMR-Egger and MVMR-IVW, generated a slightly biased estimate of $\boldsymbol{b}$ and were unable to completely eliminate collider bias for a few SNPs, especially when $p_2$ = 4. Note that, if the variance estimator (14), instead of the correct and default one, was used for MVMR-cML, the results were similar (shown in Table L and Table M in S1 Text). When 50% invalid IVs were used, some methods, such as MVMR-cML, tended to underestimate the standard errors of $\hat{\boldsymbol{b}}$ [14], and hence the standard errors of the bias-corrected estimator $\hat{\beta}_{G_iY}$, as shown in Table Y in S1 Text. In the scenario without pleiotropy (Section C in S1 Text), a small p-value $5e − 8$ was utilized for IV selection, thus only valid IVs were employed for estimation. Consequently, all MVMR methods performed equally well and successfully mitigated collider bias. Following the reviewers' suggestions, we have included two additional simulations in Section D and Section E in S1 Text. The first simulation illustrates a "null" scenario without collider bias, wherein the mean effect estimates remained unchanged after bias correction. The second simulation demonstrates the inadequacy of UMVR in mitigating the collider bias induced by two covariates.

## Applications

We applied our bias-correction approach to two previous GWAS applications. In the first, we examined a GWAS of waist-to-hip ratio (WHR) adjusted for body mass index (BMI) [11]. In the second, we considered a GWAS of BMI, utilizing principal components (PCs) of metabolomic variables as heritable covariates [5]. The GWAS results were derived from individual-level UK Biobank (UKB) data using the same subsample. By applying our bias-correction method to these GWAS applications, we aimed to assess the impact of collider bias and thus enhance the reliability of estimated SNP effects.

We followed the same data cleaning procedure described previously [5]. The analysis was performed on a dataset comprising self-reported, unrelated White individuals. For the

genotype data, we first removed individuals who were identified as outliers for heterozygosity or had a high rate missing data. Additionally, individuals with abnormal numbers of sex chromosomes and those with inconsistent self-reported sex and genetic sex were also excluded. Next, genetic variants with a minor allele frequency less than 0.01, a missing genotype rate exceeding 0.05, or failing the Hardy-Weinberg equilibrium test at a p-value threshold of $1e − 6$ were removed. After the data cleaning process, missing values were imputed using its mean value. Following the data cleaning steps, the principal components (PCs) for the 249 metabolomic biomarkers were calculated for the GWAS of BMI. The resulting dataset comprised approximately 500,000 SNPs and 100,000 individuals. All features were standardized prior to the GWAS. Moreover, the two phenotypes, WHR and BMI, underwent inverse-rank normalization to facilitate analysis [5]. For BMI, following previous study [5], significant SNPs were identified using the conventional p-value threshold of $5e − 8$ and mapped to 1,703 independent linkage disequilibrium (LD) blocks [25], with each treated as an independent genomic locus.

To select (approximately) independent IVs, we applied a pruning procedure using a window size of 50 SNPs and a linkage disequilibrium (LD) threshold of $r^2 = 0.001$. From the remaining SNPs, we selected relevant IVs for covariates based on a significance threshold of $5e − 10$. We then obtained $\hat{\boldsymbol{b}}$ using different MVMR methods and performed the bias correction for each SNP. Initially different methods gave different results. We performed a sensitivity analysis by leaving one IV out each time, some influential SNPs are identified, most of which were identified as invalid IVs by MVMR-cML. After removing these influential IVs, most or all MVMR methods gave consistent results.

## GWAS of WHR with adjustment for BMI

In this section, we applied the bias-correction methods to a GWAS of WHR, adjusting for BMI as a single covariate [26]. The previous analysis of WHR utilized BMI as a covariate with the aim of uncovering SNPs influencing WHR through pathways distinct from those mediated by BMI [26]. However, it is important to note that this approach might introduce collider bias in the estimation of the direct effect of a SNP on WHR, primarily due to the heritability of BMI [6].

To obtain the GWAS summary data of WHR, we conducted a regression analysis using the equation in (4). In our analysis, the covariate vector $\boldsymbol{J} = (J_1, \cdots, J_{p_1})^T$ consisted of sex, age, and the top 10 genetic PCs provided by UKB [5]. The vector $\boldsymbol{H}$ became a scalar of BMI. The error term was denoted as $\epsilon$. It was important to note that the conditional effects $\boldsymbol{\beta}'_{JY}$ and $\boldsymbol{\beta}'_{HY}$ could also be biased compared to their true values $\boldsymbol{\beta}_{JY}$ and $\boldsymbol{\beta}_{HY}$, respectively, due to the unmeasured confounder $U$. As our bias-correction approach requires GWAS summary data of $\boldsymbol{H}$, in this part, we also conducted GWAS of BMI (with adjustment for $\boldsymbol{J}$).

Table 3 provides the estimate $\hat{\boldsymbol{b}}$ (a scalar), its standard error (SE), and the numbers of significant SNPs and loci produced by each method. For DHO, the "Hedges-Olkin" method was employed to reduce regression dilution [7], and the SE was estimated through data perturbation. With the exception of Egger regression, all methods yielded consistent results, where $\hat{\boldsymbol{b}}$ was negative. We suspect that the positive value reported by Egger was incorrect for two reasons. First, Egger regression is not robust to correlated and directional pleiotropy [20], which might persist even after the removal of possibly invalid IVs. Second, the SE provided by Egger regression was extremely large compared to those of other methods, suggesting a substantial degree of uncertainty in the corresponding estimate. Therefore, we accepted the values given

**Table 3. Point estimates of *b* and number of significant loci obtained by applying different MVMR methods.** Standard errors (SE) are given in parenthesis.

|  | No correction | MVMR-cML | MVMR-Egger | MVMR-IVW | MVMR-Median | MVMR-Lasso | SH | DHO |
|---|---|---|---|---|---|---|---|---|
| $\hat{b}$ | *NA* | −0.079 | 0.047 | −0.058 | −0.063 | −0.063 | −0.11 | −0.068 |
| (SE) | *NA* | (0.001) | (0.322) | (0.003) | (0.004) | (0.003) | (0.002) | (0.013) |
| # of significant SNPs | 315 | 269 | 11 | 260 | 260 | 260 | 209 | 229 |
| # of significant loci | 47 | 42 | 2 | 42 | 42 | 42 | 37 | 39 |

by other methods. The estimates of *b* aligned with those in the literature, where most MVMR approaches yielded a negative value close to 0 [11].

In Table 3, fewer loci were identified after correcting collider bias. For the 315 significant SNPs before adjustment, Fig 4 shows that their effect estimates did not change significantly after bias correction, suggesting that adjusting for BMI as a covariate did not introduce severe bias. The decrease in the number of significant SNPs was mainly due to the slightly inflated variances of the effect estimates. This conclusion is consistent with a previous study showing that adjusting for BMI in a GWAS of WHR introduced only minor collider bias [11]. For Egger regression, due to the large SE of $\hat{b}$, the variance of $\hat{\beta}_{G_i Y}$ was severely inflated after bias correction, leading to a dramatic decrease in power. Only 2 significant loci were identified.

Fig 4 compares the SNP effect estimates before and after bias correction, along with their corresponding standard errors (SEs) as vertical or horizontal bars. Here we only include figures for MVMR-cML since all MVMR methods yielded similar figures; the complete set of results can be found in the Section G.1 in S1 Text.

The Manhattan plots in Fig 5 depict the significant loci before and after MVMR-cML bias correction. The upper/lower panel represents the results after/before bias correction. The

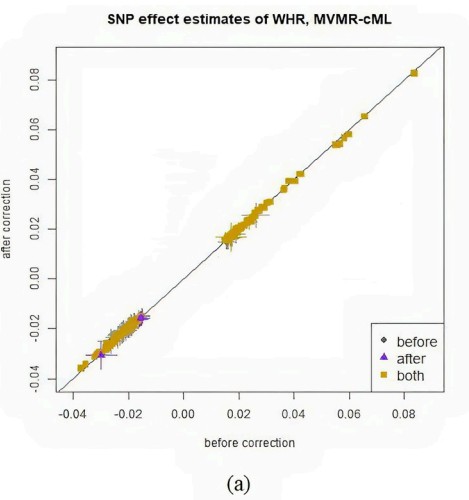
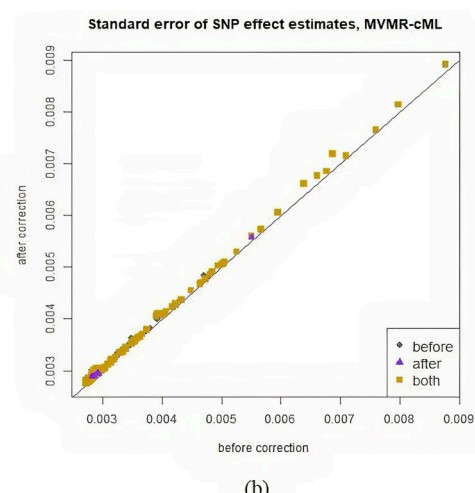

(a)　　　　　　　　　　　　　　　　　　(b)

**Fig 4. Effect estimates of WHR before and after bias correction.** Horizontal and vertical bars represent 1 SE of an estimate before and after correction respectively. SEs are given in the right column. In the legends, "before" refers to the SNPs that were significant only before bias correction, "after" refers to the SNPs significant only after bias correction, "both" refers to the SNPs significant both before and after bias correction.

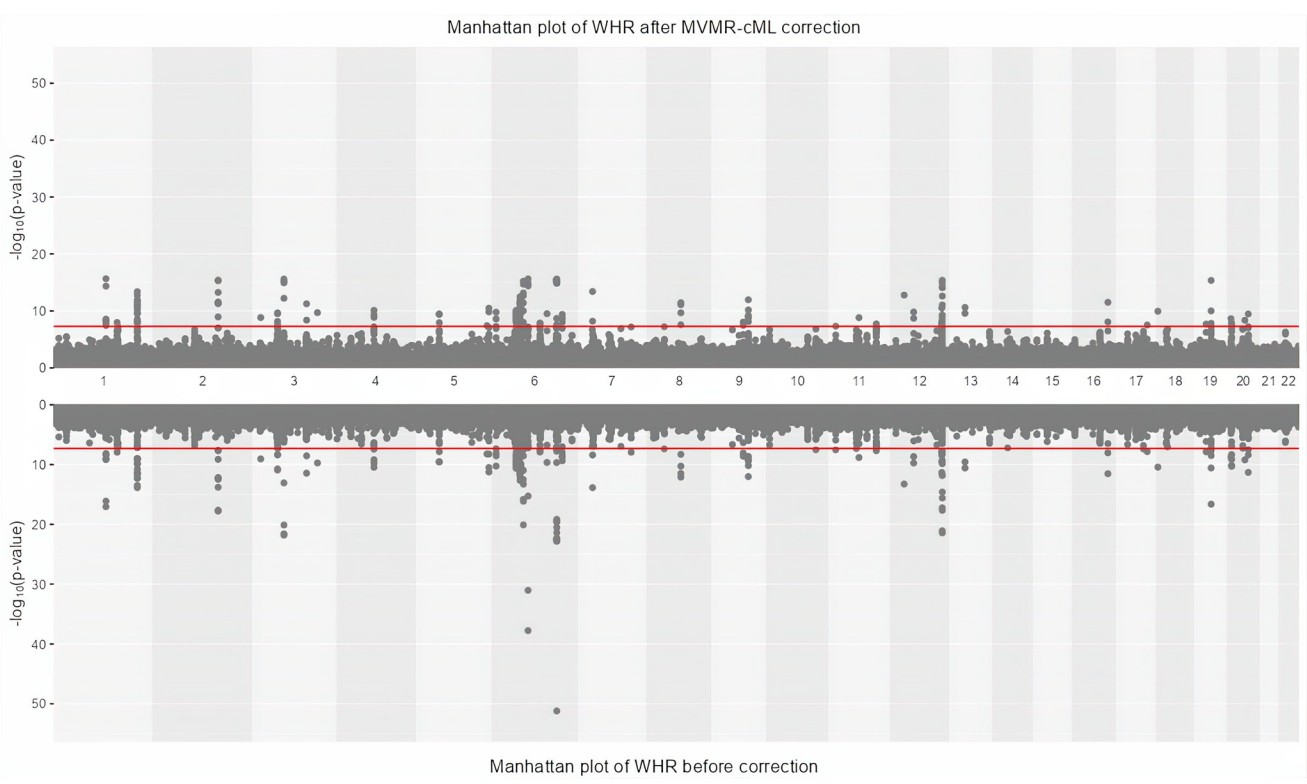

**Fig 5. Manhattan plot of WHR before (upper panel) and after (lower panel) applying MVMR-cML-bias-correction.**

overall profiles of significant SNPs/loci before and after correction are similar, but fewer SNPs remain significant after correction.

## GWAS of BMI with adjustment for metabolomic PCs

We obtained the GWAS result of BMI using Eq (4) [5]. The vector $H$ contained the principal components (PCs) derived from the 249 metabolomic variables. We employed PCs for dimension reduction since many of the metabolites exhibit high correlations [5]. $J$ and $\epsilon$ were the same as in the previous example. This regression model for BMI was referred to as **model 1 ($M_1$)** previously [5] and subsequently. To correct potential collider bias, the GWAS of metabolomic PCs was conducted using $J$ as covariates.

Instead of seeking loci that act on BMI not through metabolic pathways, the previous study aimed to enhance the power of discovering marginal SNP effects by using omic data to account for environmental/residual variation. However, it is crucial to note that including metabolomic PCs can potentially introduce collider bias for marginal effects, given both genetic and

**Table 4. The proportion of the total variance explained by and the number of SNPs significantly associated with each metabolomic PC.**

| metabolomic PC | PC1 | PC2 | PC3 | PC4 | PC5 | PC6 | PC7 | PC8 | PC9 | PC10 |
|---|---|---|---|---|---|---|---|---|---|---|
| proportion of variance | 12% | 10% | 7% | 5% | 3% | 3% | 3% | 3% | 2% | 2% |
| # of associated SNPs | 680 | 595 | 668 | 497 | 316 | 200 | 304 | 297 | 338 | 182 |

**Table 5. Point estimates of $b$ when applying different MVMR methods on $M_1$.** Standard errors (SE) are given in parenthesis.

| Dimension of $b$ | 1 | 2 | | 3 | | | 4 | | | | 5 | | | | |
|---|---|---|---|---|---|---|---|---|---|---|---|---|---|---|---|
| $\hat{b}$ | $\hat{b}_1$ | $\hat{b}_1$ | $\hat{b}_2$ | $\hat{b}_1$ | $\hat{b}_2$ | $\hat{b}_3$ | $\hat{b}_1$ | $\hat{b}_2$ | $\hat{b}_3$ | $\hat{b}_4$ | $\hat{b}_1$ | $\hat{b}_2$ | $\hat{b}_3$ | $\hat{b}_4$ | $\hat{b}_5$ |
| MVMR-cML | 0.23 | 0.24 | −0.17 | 0.24 | −0.25 | 0.16 | 0.28 | −0.35 | 0.16 | 0.15 | 0.25 | −0.29 | 0.14 | 0.22 | 0.16 |
| (SE) | (0.03) | (0.03) | (0.05) | (0.03) | (0.05) | (0.03) | (0.03) | (0.07) | (0.03) | (0.04) | (0.04) | (0.06) | (0.03) | (0.04) | (0.04) |
| MVMR-Egger | 0.16 | 0.22 | −0.14 | 0.28 | −0.20 | 0.16 | 0.24 | −0.27 | 0.14 | 0.16 | 0.23 | −0.23 | 0.13 | 0.22 | 0.16 |
| (SE) | (0.08) | (0.06) | (0.05) | (0.05) | (0.05) | (0.03) | (0.06) | (0.06) | (0.03) | (0.04) | (0.05) | (0.06) | (0.03) | (0.04) | (0.04) |
| MVMR-IVW | 0.23 | 0.23 | −0.14 | 0.23 | −0.21 | 0.16 | 0.26 | −0.27 | 0.14 | 0.17 | 0.23 | −0.23 | 0.13 | 0.22 | 0.16 |
| (SE) | (0.03) | (0.03) | (0.05) | 0.03 | 0.05 | 0.03 | (0.04) | (0.06) | (0.03) | (0.04) | (0.03) | (0.06) | (0.02) | (0.04) | (0.04) |
| MVMR-Lasso | 0.26 | 0.23 | −0.14 | 0.23 | −0.21 | 0.16 | 0.26 | −0.27 | 0.14 | 0.17 | 0.24 | −0.24 | 0.13 | 0.23 | 0.16 |
| (SE) | (0.04) | (0.03) | (0.05) | (0.03) | (0.05) | (0.03) | (0.04) | (0.06) | (0.03) | (0.04) | (0.03) | (0.06) | (0.02) | (0.04) | (0.04) |
| MVMR-Median | 0.23 | 0.23 | −0.12 | 0.23 | −0.18 | 0.15 | 0.26 | −0.25 | 0.16 | 0.15 | 0.23 | −0.20 | 0.14 | 0.19 | 0.15 |
| (SE) | (0.03) | (0.04) | (0.08) | (0.04) | (0.08) | (0.05) | (0.05) | (0.10) | (0.05) | (0.06) | (0.05) | (0.09) | (0.04) | (0.06) | (0.05) |
| DHO | 0.16 | NA | | NA | | | NA | | | | NA | | | | |
| (SE) | (0.01) | | | | | | | | | | | | | | |
| SH | 0.30 | | | | | | | | | | | | | | |
| (SE) | (0.04) | | | | | | | | | | | | | | |

environmental influences on metabolomic measurements. Table 4 presents the number of associated SNPs for the top 10 PCs at the genome-wide significance level of $5e − 8$. To mitigate collider bias, we applied our bias-correction approach and compared the results with previous findings. Specifically, we explored scenarios where the dimension of the vector $\boldsymbol{H}$, denoted as $p_2$, varied from 1 to 5. In other words, we first fitted $M_1$ with the top $p_2$ metabolomic PCs and then adjusted possible collider bias induced by these $p_2$ covariates using different MVMR methods. This allowed us to examine the results of including up to 5 metabolomic PCs in the analysis. Additionally, as suggested by previous research [5], we also provided results using the top 20 metabolomic PCs in Section G.2 in S1 Text.

To validate the identified loci for BMI, we followed a previously established validation approach [5] and utilized two GWAS summary datasets as validation data based on their larger sample sizes. Specifically, we used the UK Biobank (UKB) GWAS round 2 results (sample size 361,194) for partial validation, which were published by the Neale lab in 2018 (http://www.nealelab.is/uk-biobank). We also incorporated the summary results from a meta-analysis of GIANT and UKB data, which included around 700,000 samples [27].

Table 5 provides the estimates of $\boldsymbol{b}$ and their standard errors obtained by applying various methods on $M_1$. Overall, the values were consistent. Therefore, different methods yielded similar estimates $\hat{\beta}_{G_iY}$ after bias correction. We performed the conditional $F$-test for weak IVs when $p_2 > 1$ [28], and all $F$-statistics were above the threshold value of 10, indicating no severe weak instrument bias. The set of complete results is available in Section G.2.2 in S1 Text.

Table 6 provides the numbers of significant SNPs and loci identified with and without bias correction. In general, we observed that the number of loci decreased after the correction. This reduction could be attributed to two main reasons. Firstly, some SNPs were identified as significant in $M_1$ due to collider bias, as certain metabolomic PCs were associated with these SNPs. For instance, when only the first metabolomic PC was involved, out of the 306 significant SNPs identified as significant in $M_1$, 99 of them became non-significant after MVMR-cML-bias-correction. Among these 99 SNPs, 58 were associated with the first metabolomic PC. Similarly, when two metabolomic PCs were included, $M_1$ identified 483 significant SNPs, but after the correction of MVMR-cML, 316 of them were no longer significant. Among these 316 SNPs, 170 were associated with at least one of the two metabolomic PCs. As $p_2$ increased,

**Table 6. Number of significant loci after applying different bias-correction methods on $M_1$.**

| # of metabolomic PCs in analysis | | No correction | MVMR-cML | MVMR-Egger | MVMR-Lasso | MVMR-Median | MVMR-IVW | DHO | SH |
|---|---|---|---|---|---|---|---|---|---|
| 1 | # of significant SNPs | 306 | 221 | 220 | 223 | 219 | 223 | 219 | 214 |
| | # of significant loci | 49 | 33 | 34 | 35 | 33 | 35 | 34 | 30 |
| | UKB validation | 45 | 33 | 34 | 35 | 33 | 35 | 34 | 30 |
| | other validation | 42 | 30 | 31 | 32 | 30 | 32 | 31 | 29 |
| 2 | # of significant SNPs | 483 | 192 | 193 | 194 | 179 | 192 | NA | NA |
| | # of significant loci | 60 | 32 | 31 | 32 | 28 | 32 | | |
| | UKB validation | 45 | 32 | 31 | 32 | 28 | 32 | | |
| | other validation | 46 | 29 | 28 | 29 | 26 | 29 | | |
| 3 | # of significant SNPs | 468 | 195 | 187 | 196 | 180 | 196 | NA | NA |
| | # of significant loci | 51 | 30 | 28 | 30 | 27 | 30 | | |
| | UKB validation | 42 | 30 | 28 | 30 | 27 | 30 | | |
| | other validation | 43 | 28 | 26 | 27 | 25 | 27 | | |
| 4 | # of significant SNPs | 553 | 195 | 191 | 190 | 174 | 190 | NA | NA |
| | # of significant loci | 62 | 29 | 28 | 27 | 26 | 27 | | |
| | UKB validation | 44 | 29 | 28 | 27 | 26 | 27 | | |
| | other validation | 46 | 27 | 25 | 25 | 24 | 25 | | |
| 5 | # of significant SNPs | 558 | 185 | 183 | 183 | 169 | 183 | NA | NA |
| | # of significant loci | 63 | 24 | 23 | 23 | 23 | 23 | | |
| | UKB validation | 45 | 14 | 24 | 22 | 23 | 22 | | |
| | other validation | 47 | 23 | 22 | 22 | 23 | 22 | | |

the number of non-significant SNPs after correction that were associated with metabolomic PCs also increased. For example, when $p_2 = 5$, out of 558 SNPs identified by $M_1$, 417 were no longer significant after the correction of MVMR-cML, and among these 417 SNPs, 267 were associated with metabolomic PCs. Secondly, any bias-correction method yielded a larger variance of $\hat{\beta}_{G_iY}$, leading to a reduction in power. On the other hand, our bias-correction method also uncovered some significant SNPs that were missed without bias correction. For instance, when 5 metabolomic PCs were included and MVMR-cML was used to obtain $\hat{b}$, 44 SNPs were identified as significant only after the bias correction. Overall, for each number of covariates, all MVMR methods yielded similar number of significant loci. In Table AC in S1 Text, we show the results when 6 to 10 metabolomic PCs were adjusted: few or no loci were identified after bias correction, possibly due to variance inflation.

Fig 6 compares the SNP effect estimates before and after bias correction, along with their corresponding standard errors as vertical or horizontal bars. We only include MVMR-cML since all MVMR methods produced similar figures. And we only present the results for the cases with $p_2 = 1$, 2, or 5 as representatives. The complete results can be found in Section G.2 in S1 Text.

Fig 6(a) shows the results when only the first metabolomic PC was included. For the SNPs that were significant only before the correction, our bias-correction approach pulled their effect estimates closer to 0, indicating that collider bias may have led to false positive results for these SNPs. Fig 6(c) and 6(e) demonstrate that as more covariates were included, a greater number of SNPs became non-significant after the correction. In each figure, many points had their effect estimates closer to 0 after the correction. For the SNPs that were significant both with and without the bias correction, their effect estimates were hardly affected by collider bias and remained the same after the correction. When only one covariate was adjusted, the SEs of $\hat{\beta}_{G_iY}$ increased only slightly after bias correction: as shown in Fig 6(b), all points are close to

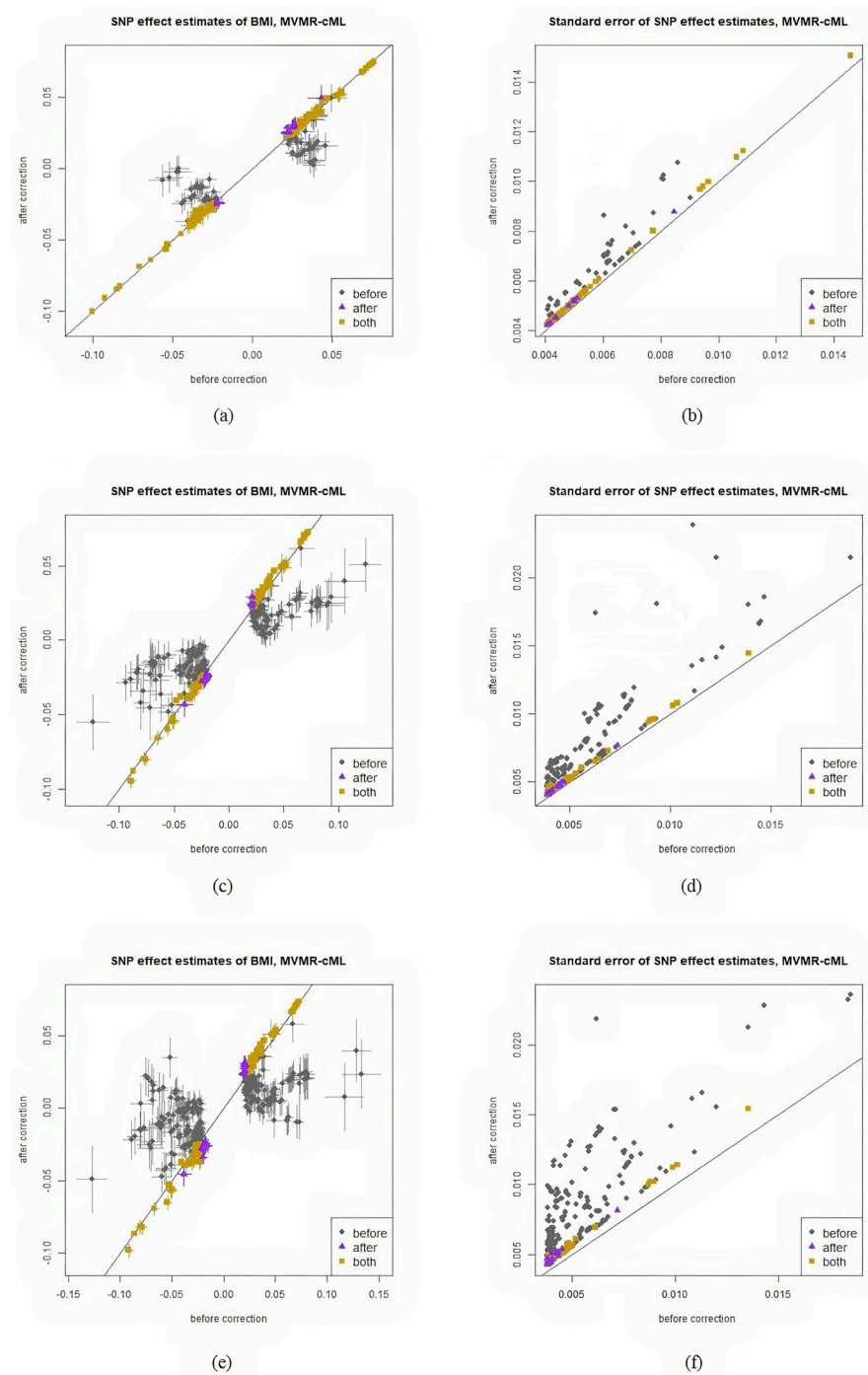

**Fig 6. Effect estimates (in $M_1$) of BMI before and after bias correction.** Horizontal and vertical bars represent 1 SE of an estimate before and after correction respectively. SEs are given in the right column. (a)-(b):1 metabolomic PC is used. (c)-(d): 2 metabolomic PCs are used. (e)-(f): 5 metabolomic PCs are used. In the legends, "before" refers to the SNPs that are significant only before bias correction, "after" refers to the SNPs that are significant only after bias correction, "both" refers to the SNPs that are significant both before and after bias correction.

the identity line. However, as more covariates/PCs were included, bias-correction led to much larger SEs.

The Manhattan plots in Fig 7 illustrate the significant loci before and after bias correction by MVMR-cML. An upper/lower panel represents the results after/before bias correction. After correction, less significant p-values were observed. As more metabolomic PCs were included, fewer significant SNPs and loci remained after bias correction.

## Results after removing genetic components of covariates

To mitigate collider bias, a strategy is to remove the genetic components from the covariates before their inclusion in GWAS analysis [5]. We removed the genetic components of the metabolomic PCs and used the corresponding residuals as the covariates in the GWAS of BMI. Then we applied the bias-correction methods, where each MVMR method yielded different estimates of $b$ due to the presence of weak instrument bias. This bias arose because, after removing genetic components, few or no SNPs were strongly associated with the covariates. The conditional $F$ (shown in Section G.2.2 in S1 Text) test also indicated the presence of weak instruments. However, it is noteworthy that different MVMR methods consistently produced similar results after bias correction. As shown in Section G.2.1 in S1 text, it was confirmed that the effect estimates in such a model were closer to each other before and after bias correction; in other words, there was no or little collider bias. On the other hand, as before, bias-correction for more covariates led to more inflated variances and possible loss of power.

## Discussion

In this study, we have addressed a general issue of collider bias in GWAS conditional analysis when multiple heritable covariates are included or adjusted. We have proposed an extension to an existing work [11], the latter of which, along with other existing ones, can handle only one covariate. Our derivation demonstrates that the estimation of the potential collider bias corresponds to a multivariable instrumental effect regression problem, allowing for the application of any valid MVMR methods. We have specifically employed MVMR-cML due to its robustness against the presence of invalid IVs violating any or all of the three valid IV assumptions [14]. More importantly here, due to its framework of constrained maximum likelihood, we can derive its various distributional characteristics, including correlations among various statistics, in the presence of overlapping samples across the GWAS of the outcome and heritable covariates. In contrast, it is unknown how to do so with other MVMR methods. Thus, our method with MVMR-cML is applicable to GWAS summary data for both the covariates and the trait of interest in a 1-sample, 2-sample or overlapping-sample setting. Through extensive simulations, we compared our method to other state-of-the-art MVMR methods, allowing for the inclusion of invalid IVs violating one of the three valid IV assumptions (i.e. Assumption A3, but not A2). The results indicated that most MVMR methods can reduce collider bias under different scenarios. These findings align with previous conclusions in the literature [11, 14], highlighting the effectiveness of our approach in eliminating or reducing collider bias. By incorporating our bias-correction approach into GWAS regression analyses, researchers can mitigate the potential bias introduced by including multiple covariates, obtaining less biased estimates of SNP-trait associations. However, as more covariates are included in the analysis, estimating and thus correcting the collider bias becomes more challenging due to increasing estimation errors and uncertainties.

In this paper, our main focus has been on correcting collider bias in conditional analyses of GWAS when adjusting for one or more heritable covariates. There are also wide-ranging applications in mediation analysis with one or more heritable covariates as potential

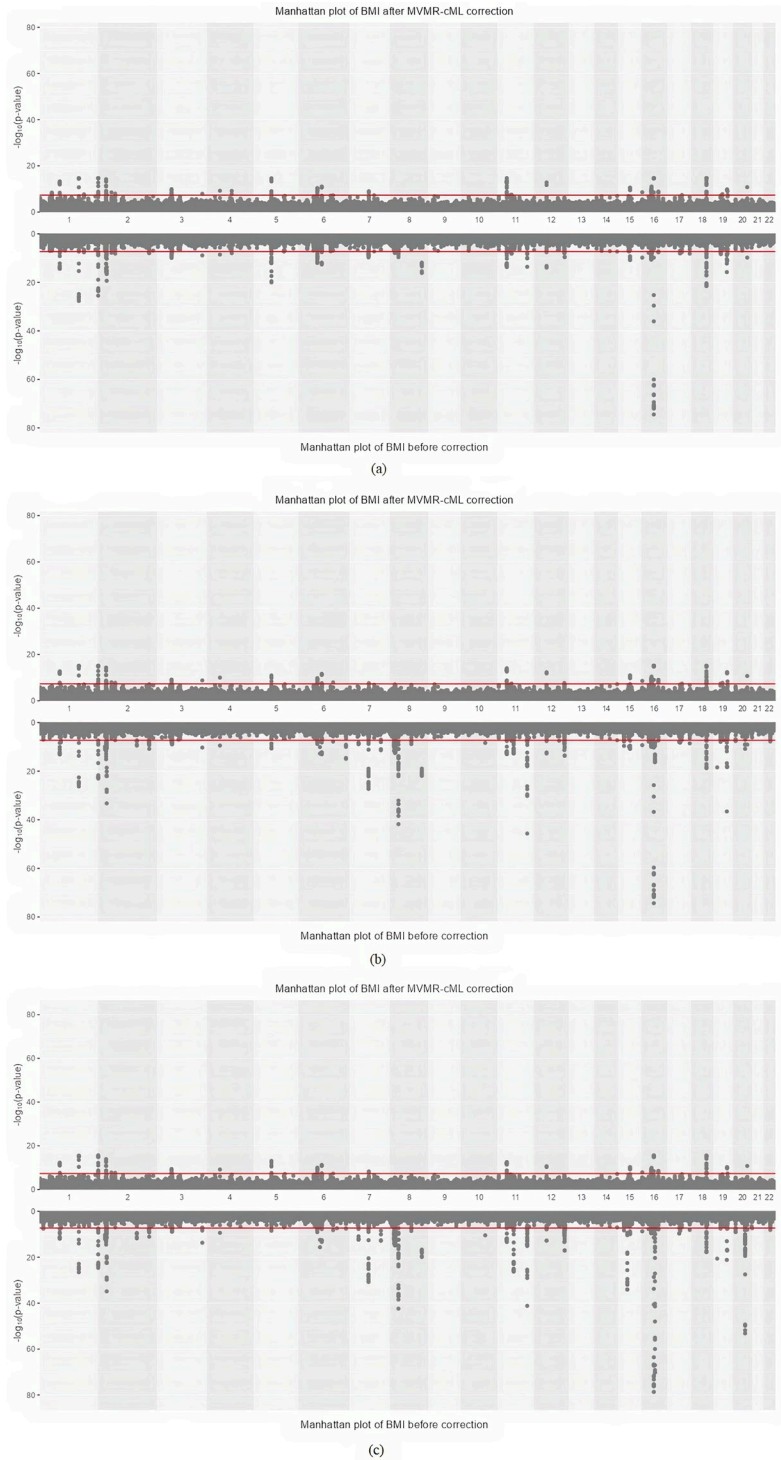

**Fig 7. Manhattan plot of BMI before (upper panel) and after (lower panel) applying MVMR-cML bias correction (in $M_1$).** (a): one metabolomic PC is adjusted, (b): two metabolomic PCs are adjusted, (c): five metabolomic PCs are adjusted.

mediators, where hidden confounding and thus collider bias are often ignored [29–38]. As mentioned by one reviewer, one limitation of our study is the restriction to linear models with quantitative traits and covariates. For binary traits, logistic regression is typically employed. While a logistic regression model can be well approximated by a linear regression model in marginal GWAS analysis of SNPs due to their small effect sizes, however, the approximation may be inadequate in conditional GWAS analysis with covariates of potentially large effects. Consequently, our bias-correction method cannot be directly applied in the latter case. It remain open how to extend our proposed bias-correction approach to accommodate binary traits. Furthermore, it is important to note that collider bias can arise in other scenarios. For instance, when the trait of interest is a precursor of, not subsequent to a heritable disease, conditioning on the disease incidence may bias the estimation of the direct SNP effects, since there are shared confounders between the trait and disease [7]. One specific example is that knowing BMI as a cause of type-2 diabetes, a SNP causing type-2 diabetes may have a biased association with BMI when studied within a case-only sample of type-2 diabetes [7]. In addition, participation in a GWAS of disease prognosis is often conditional on survival until time of recruitment, and possibly other health conditions. But there may be unknown common causes of survival and prognosis that create further biases [7]. It would be useful to extend our method to investigate these problems.

## Supporting information

**S1 Text. Supplementary file with theory and derivations/proofs, more simulation results and additional real data analysis results.**
(PDF)

## Acknowledgments

This study was supported by the Minnesota Supercomputing Institute at the University of Minnesota.

## Author Contributions

**Conceptualization:** Wei Pan.

**Data curation:** Peiyao Wang.

**Formal analysis:** Peiyao Wang, Zhaotong Lin, Haoran Xue, Wei Pan.

**Funding acquisition:** Wei Pan.

**Investigation:** Peiyao Wang, Zhaotong Lin, Haoran Xue, Wei Pan.

**Methodology:** Peiyao Wang, Zhaotong Lin, Haoran Xue, Wei Pan.

**Project administration:** Wei Pan.

**Resources:** Wei Pan.

**Software:** Peiyao Wang, Zhaotong Lin.

**Supervision:** Wei Pan.

**Validation:** Peiyao Wang.

**Visualization:** Peiyao Wang.

**Writing – original draft:** Peiyao Wang.

**Writing – review & editing:** Zhaotong Lin, Haoran Xue, Wei Pan.

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
