## [Decision Letter · Decision Letter 0]

25 Jan 2024

Dear Dr Pan,

Thank you very much for submitting your Research Article entitled 'Collider Bias Correction for Multiple Covariates in GWAS Using Robust Multivariable Mendelian Randomization' to PLOS Genetics.

The manuscript was fully evaluated at the editorial level and by independent peer reviewers. The reviewers appreciated the attention to an important problem, but raised some substantial concerns about the current manuscript. Based on the reviews, we will not be able to accept this version of the manuscript, but we would be willing to review a much-revised version. We cannot, of course, promise publication at that time.

If you decide to revise the manuscript for further consideration at PLOS Genetics, please aim to resubmit within the next 60 days, unless it will take extra time to address the concerns of the reviewers, in which case we would appreciate an expected resubmission date by email to plosgenetics@plos.org.

We are sorry that we cannot be more positive about your manuscript at this stage. Please do not hesitate to contact us if you have any concerns or questions.

Yours sincerely,

Xiang Zhou, Ph.D.

Academic Editor

PLOS Genetics

Michael Epstein

Section Editor

PLOS Genetics

Reviewer's Responses to Questions

**Comments to the Authors:**

Reviewer #1: The manuscript outlines a novel approach for estimating collider bias in GWAS, utilizing a robust multivariable Mendelian Randomization (MVMR) technique that's based on constrained maximum likelihood, referred to as MVMR-cML. The objective of MVMR-cML is to infer the causal effects of multiple exposures on the outcome in scenarios where the instrumental variables (IVs) are invalid and correlated pleiotropy exists. I have some concerns that are listed below to further enhance the manuscript.

1. In Figure 2, H1 is both the mediator in the pathway G→H1→Y and a collider in the pathway G→H1←U→Y. I wonder how this can be determined in practice, that is, how can we test whether H1 plays both roles as those depicted in Figure 2?

2. I suggested the authors conducted additional simulations to compare the performance of the traditional GWAS and the proposed bias correction method when H is not a collider?

3. If H1 is a mediator, it should not be accounted for in GWAS analysis. In addition, if we remove the edge H→Y and render H1 a complete collider, it is generally not recommended adjust for it in GWAS. Thus, Figure 2 is confusing.

4. In Figure 4, the effect estimates of WHR before and after bias correction appear to be quite similar. I suggested the authors to try different data set to highlight the advantage of the proposed method.

5. The manuscript attempts to utilize MVMR to correct collider bias in GWAS, but the depiction of direct effect or indirect effect in Figure 2 is confusing.

6. The symbols in the manuscript are highly confusing and most of them are not explained, which seriously impacts the readability of the article. For example, in formula (3), K represents variables, but in formula (9), K denotes the number of invalid IVs. Furthermore, in section 2.5, rho represents correlation, whereas in section 3.1, rho denotes the proportion of invalid instrumental variables. What does the sudden appearance of V in formula (1) represent? What are its practical implications? In the simulation settings, V disappears again. Is V a vector of covariates or SNPs?

7. If Y is a binary variable, such as diseases, does this method remain effective?

8. MVMR assumes the IV is associated with at least one of H, and the SNP directly associated with Y is invalid SNP. GWAS aims to find the SNPs directly associated with Y, but MVMR-cML assume the plurality condition, and in formula (9) the number of invalid IV is constrained to K. How to interpret this?

9. How to use this method in practice? Did this method be applied to thousands of SNPs one by one? What is the computational time?

10. I strongly suggested the authors to re-organize the introduction of the manuscript, to further highlight the significance and motivation of the proposed method.

Reviewer #2: This paper develops instrument variables correction for collider bias when there are multiple colliders, such as GWAS when conditioning on multiple heritable covariates. This extends earlier work that corrected for a single collider such as disease incidence. The authors show that it works well in conjunction with their own MVMR-cML approach for multivariate Mendelian randomisation. This is a good contribution to the literature. I have the following comments.

1. In several places the authors state that their method can account for overlapping samples, and even say (section 2.4) that they have simulations confirming that previous methods have inflated type-1 errors in this case. However I could find no such simulations in the paper. Moreover, Dudbridge et al (2019) argued that their approach was robust to overlapping samples, and confirmed it in their simulations. This was further established by Barry et al. (PLoS Genet 2021). Please explain how your situation differs from theirs, and provide simulations to back up any new claims.

2. The authors claim to have derived the variance of their estimator, and that this goes beyond previous work. However, across all the results the new method’s SE is consistently overestimated compared to the empirical SD, and the type-1 errors are conservative, whereas previous methods appear to be well calibrated. So something isn’t right here.

3. In some places the superscript T means “transpose”, and in others “total” – this is confusing. See eg equations 2-3 and the text above.

4. P9 “The correlation parameters in \\Sigma_j…” repeats earlier text.

5. The simulation used 1000 SNPs from chromosome 6. Why was this chromosome used, and was the HLA region avoided?

6. Section 3.1.1, I couldn’t understand why the type-1 errors were not correctly controlled, not just “better controlled”, especially at p2=4. I see that the simulation had invalid IVs, but the pleiotropy appears to be balanced with the InSIDE assumption valid, so the methods should all be OK. What is happening here? It would help to give a simulation under the assumptions of the method, showing that it does work in that case.

7. I was surprised that the authors have considered Egger regression, since the same authors have published a convincing refutation of that method.

8. P15 “This deviation was attributed to … more covariates”. This seems to attribute bias to an increase in sampling variance, which isn’t right.

9. Supp P6. Second line of proof, “negative likelihood” should be log likelihood.

Reviewer #3: The authors proposed using an MVMR approach (MVMR-cML) that they recently developed to perform collider bias correction in obtaining GWAS summary statistics. I think the idea is interesting and can potentially be more reliable than a univariate MR approach. However, I also have some concerns related to the setup of the framework, method details and data evaluation. Here are my major comments:

1. Foundation of the causal framework setup. In Figure 2, the authors draw the causal DAG and claims that the true parameter to estimate should be the direct effect of SNP j \\beta_{G_iY}. I think this is very confusing and misleading as \\beta_{G_iY} will never be the direct effect of SNP j even when there is no collider bias as other SNPs are not jointly considered. Even without collider bias, the marginal associations between SNP j and Y without adjusting for other SNPs are never claimed to be any causal effect of SNP j. Figure 2 is also not qualified to be a DAG as only a single SNP is considered here. Given that what GWAS summary statistics provide are never any causal effect of SNP j, I think the target of the problem in the paper to adjust for collider bias is very vague. I do agree that collider bias exists but I'm only convinced that it should be adjusted when we want to identify causal SNPs using methods that jointly consider multiple SNP such as SUSIE. I'm not convinced for GWAS summary statistics. Thus, I think the authors should clarify the setup and maybe give a few more concrete examples to address the necessity.

2. Another problem for the DAG in Figure 2 is that covariates such as population ancestry (maybe denoted as V in the paper?) that are confounders of U and G are not included. In the structural equations (1) the authors simplify that V is not correlated with U nor G. If V are population ancestries, this assumption will not be true and I'm not sure whether it will affect the subsequent calculations or not.

3. For the MVMR-cML method the authors claim that in equation (9), the number of invalid IVs K can be as large as l-p_2 -1. I'm not sure if this is possible. I think the slope b in (5) would not be identifiable if beta_{G_iY} is not sparse enough. I hope the authors can clarify more.

4. Since the authors need to work with individual GWAS data anyway, what is the benefit of using summary GWAS MR?

5. Related to the previous question, if the trait is binary, than a linear model might not be appropriate, is the method proposed in the paper still applicable?

6. To make the paper more convincing, I think the authors need to have some real data example to illustrate that univariate MR is not enough in practice

A minor comment:

Section 2.1 the authors briefly mentioned mtCOJO. I think that part is very hard to follow without reading the mtCOJO paper and the authors should make the section more self-contained.

**Have all data underlying the figures and results presented in the manuscript been provided?**

Reviewer #1: None

Reviewer #2: None

Reviewer #3: Yes

PLOS authors have the option to publish the peer review history of their article (what does this mean?). If published, this will include your full peer review and any attached files.

Reviewer #1: No

Reviewer #2: **Yes: **Frank Dudbridge

Reviewer #3: No

---

## [Decision Letter · Decision Letter 1]

17 Mar 2024

Dear Dr Pan,

Thank you very much for submitting your Research Article entitled 'Collider Bias Correction for Multiple Covariates in GWAS Using Robust Multivariable Mendelian Randomization' to PLOS Genetics.

The manuscript was fully evaluated at the editorial level and by independent peer reviewers. The reviewers appreciated the attention to an important topic but identified some concerns that we ask you address in a revised manuscript.

We therefore ask you to modify the manuscript according to the review recommendations. Your revisions should address the specific points made by each reviewer.

Yours sincerely,

Xiang Zhou, Ph.D.

Academic Editor

PLOS Genetics

Michael Epstein

Section Editor

PLOS Genetics

Reviewer's Responses to Questions

**Comments to the Authors:**

Reviewer #1: The authors hava addressed all my comments.

Reviewer #2: The authors responded to my comments, and I now understand the paper better. Thank you.

There are a number of typos in the new text, and some remaining in the old. The most serious of which is "casually" instead of "causally".

Reviewer #3: The revised paper has resolved most of my previous concerns. I only have two remaining comments.

1. Foundation of the casual framework setup. I appreciate authors' effort in providing better description of the setup, but frankly speaking, it is still very confusing. The description now is a mixture of causal inference language and associations where the definitions are not coherent. If the direct effects that the authors care about are just conditional associations of a single SNP G on Y conditional on X, which has a clear definition by its own, seems that the casual concepts like mediation and collider bias, and the DAG in Fig1 are not relevant. The authors have also created new terms like direct association, indirect associations and spurious associations, which I don't think are clearly defined concepts and can be confusing. While the authors change from causal effects to different types of associations in the text, they still call models (1)-(3) on a single SNP as the "true causal model for each SNP G_i". These descriptions seem to be conflicting with each other.

I feel that it is possible that the authors describe the problem clearly by starting with a causal model considering all SNPs, discuss the collider bias the mediation in that model and then discuss how it affects the single-SNP associations. This might be beyond the scope of the paper. The authors may add the limitation of their framework in the discussion section.

2. For my previous point 3, I did not understand the authors responses. Does the authors mean that b is still identifiable when "invalid IVs K can be as large as l − p2 − 1"? How is this relavant to the "multivariable plurality condition"?

**Have all data underlying the figures and results presented in the manuscript been provided?**

Reviewer #1: None

Reviewer #2: None

Reviewer #3: None

PLOS authors have the option to publish the peer review history of their article (what does this mean?). If published, this will include your full peer review and any attached files.

Reviewer #1: No

Reviewer #2: **Yes: **Frank Dudbridge

Reviewer #3: No

---

## [Editor Report · Decision Letter 2]

2 Apr 2024

Dear Dr Pan,

We are pleased to inform you that your manuscript entitled "Collider Bias Correction for Multiple Covariates in GWAS Using Robust Multivariable Mendelian Randomization" has been editorially accepted for publication in PLOS Genetics. Congratulations!

Yours sincerely,

Xiang Zhou, Ph.D.

Academic Editor

PLOS Genetics

Michael Epstein

Section Editor

PLOS Genetics

Comments from the reviewers (if applicable):

**Data Deposition**

http://datadryad.org/submit?journalID=pgenetics&manu=PGENETICS-D-23-01347R2

**Press Queries**

---

## [Editor Report · Acceptance letter]

16 Apr 2024

PGENETICS-D-23-01347R2 

Collider Bias Correction for Multiple Covariates in GWAS Using Robust Multivariable Mendelian Randomization 

Dear Dr Pan, 

We are pleased to inform you that your manuscript entitled "Collider Bias Correction for Multiple Covariates in GWAS Using Robust Multivariable Mendelian Randomization" has been formally accepted for publication in PLOS Genetics! Your manuscript is now with our production department and you will be notified of the publication date in due course.

With kind regards,

Anita Estes

PLOS Genetics

On behalf of:
